# Group Robust Preference Optimization in Reward-free RLHF

**Shyam Sundhar Ramesh**[1]
University College London (UCL)

**Yifan Hu**
ETH Zurich, EPFL

**Iason Chaimalas**
University College London (UCL)

**Viraj Mehta**
TensorZero

**Pier Giuseppe Sessa**
ETH Zurich

**Haitham Bou Ammar**
University College London (UCL)
Huawei Noah's Ark Lab

**Ilija Bogunovic**
University College London (UCL)

## Abstract

Adapting large language models (LLMs) for specific tasks usually involves fine-tuning through reinforcement learning with human feedback (RLHF) on preference data. While these data often come from diverse labelers' groups (e.g., different demographics, ethnicities, company teams, etc.), traditional RLHF approaches adopt a "one-size-fits-all" approach, i.e., they indiscriminately assume and optimize a single preference model, thus not being robust to unique characteristics and needs of the various groups. To address this limitation, we propose a novel Group Robust Preference Optimization (GRPO) method to align LLMs to individual groups' preferences robustly. Our approach builds upon reward-free direct preference optimization methods, but unlike previous approaches, it seeks a robust policy which maximizes the worst-case group performance. To achieve this, GRPO adaptively and sequentially weights the importance of different groups, prioritizing groups with worse cumulative loss. We theoretically study the feasibility of GRPO and analyze its convergence for the log-linear policy class. By fine-tuning LLMs with GRPO using diverse group-based global opinion data, we significantly improved performance for the worst-performing groups, reduced loss imbalances across groups, and improved probability accuracies compared to non-robust baselines.

## 1 Introduction

As the usage of large language models (LLMs) has grown in recent years, the question of their *alignment* has come to the forefront. Their remarkable capability to address a wide range of tasks (Radford et al. [36]) stems from pre-training on a self-supervised objective over internet-scale text. This vast internet-scale content, however, carries a higher risk of biases, inaccuracies, and controversial content than smaller, curated datasets. Thus, ensuring that the model's responses and behaviors correspond to human intentions and values is crucial.

Typical approaches to alignment [11, 34, 37] involve gathering preference feedback from human labelers to train models that reflect their desires. Such approaches often treat individual preferences as samples from a broader preference distribution. However, this perspective often oversimplifies the complex reality that human societies consist of numerous *distinct groups* (e.g., different demographics, ethnicities, company teams, etc.), each with their own set of preferences that can significantly

---

[1] Corresponding author: `shyam.ramesh.22@ucl.ac.uk`.

38th Conference on Neural Information Processing Systems (NeurIPS 2024).

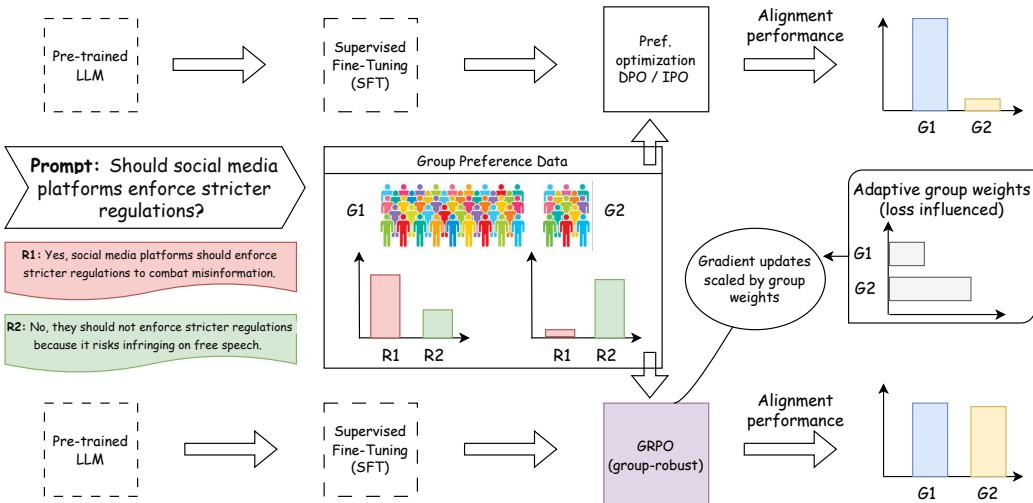

Figure 1: Current reward-free preference optimization methods typically optimize based on average human feedback. This often aligns predominantly with the preferences of the majority group (G1, R1 > R2) at the expense of minority groups (G2, R2 > R1). In contrast, our GRPO algorithm introduces adaptive weighting for different user groups and prioritizes optimizing for the worst-case group performance, leading to better alignment for the most disadvantaged groups.

diverge. Consequently, prevalent alignment strategies tend to adopt a "one-size-fits-all" model and disproportionately favor the preferences of the majority group, often at the expense of minority groups and their preferences, as illustrated in Figure 1.

To improve alignment performance for even the most disadvantaged groups, we propose to robustly solve the problem of diverse group preferences by (i) including group information in the context of the LLM and (ii) optimizing against the *worst-case* alignment performance across all groups. We develop policies that guarantee equitable performance across all groups, ensuring that no group is disproportionately disadvantaged due to inherent biases or imbalances in the training data.

**Related work.** The established process for alignment of LLMs using Reinforcement Learning from Human Feedback (RLHF) is set out in [45, 60] and [34]. The RLHF fine-tuning process consists of learning a reward model from human comparisons between responses to a given prompt, using the Bradley-Terry model [5]. Then, one performs policy optimization using Proximal Policy Optimization [40] to learn a policy that maximizes the learned reward function. For a comprehensive overview and perspective of the RLHF topic, we refer the reader to [24, 7, 23].

Due to the challenges of tuning PPO and the vulnerability of reward models ([50, 37, 17, 49]), alternative approaches to PPO-based RLHF have been proposed, including rejection sampling fine-tuning [14, 18, 49, 30] and conditional supervised fine-tuning [20, 54, 9]. In particular, Rafailov et al. [37] introduce Direct Preference Optimization (DPO), which optimizes policies directly based on human preferences, avoiding the need for a separate reward model. This approach simplifies training and reduces reward overfitting. Other studies, such as [1, 58, 47, 43, 16], propose novel *reward-free* RLHF methods, with some bypassing preference datasets altogether ([16, 6]). We utilize a reward-free framework similar to [37, 1], however, unlike previous works that assume a single preference distribution, we consider multiple preference distributions from diverse groups. Further, we aim to robustly fine-tune the LLM to ensure minimal disparity in performance across all groups. Other studies addressing robustness in preference optimization include [21] and [26]. However, these works primarily focus on different aspects of robustness, such as robustness to noise and resilience against out-of-preference data.

Robust language modeling techniques have been studied by [33, 52] to optimize performance of language models over a wide-range of topics. They consider robust pre-training of language models based on the group Distributionally Robust Optimization (DRO) approach. A concrete theoretical study of the group DRO approach was performed in [39] and applied to vision problems. These are designed by extending previous minimax algorithm for solving DRO from [31]. In the RLHF setup, [3] consider weighting of loss from different topics (harmless vs helpful) for robust reward learning. Also, Chakraborty et al. [8] consider robust policy optimization by learning multiple reward functions corresponding to sub-populations and learning a robust policy w.r.t. the learned

rewards. Differing from these works, we embed group robustness directly into the reward-free tuning paradigm. We provide an concrete algorithm that adaptively weighs the loss for different groups and optimizes for a policy that minimizes the weighted loss. Further, our algorithm employs a novel gradient estimator tailored to the group robust DPO problem.

In the non-robust setup, [57] also explore group-based preference learning with LLMs by including group information in prompts with a transformer module that is trained optimally to choose an example sequence of prompts, LLM responses, and group preferences for in-context learning. [49] consider alignment with user preferences assuming that each user has varied importance over the distinct metrics in their multi-objective reward model. In contrast, we are not modeling multi-reward objectives but consider a reward-free setting. And, our methodology directly models each group's preferences through a group-dependent latent reward model where the group dependency is injected through the prompt. Further, unlike the non-robust problem setups of [57, 49], we consider the robust alignment problem optimizing for the worst group performance. We detail other related works extensively in Appendix A.

**Main Contributions.** The following are the main contributions of this work: (i) We present GRPO, the group robust formulation of Direct Preference Optimization (DPO) [37], wherein we augment the context of the LLM with the group information, and pose the problem as a robust optimization problem to minimize the worst-case loss amongst the diverse groups. We also show that a naive application of group robustness to the LLM policy maximization objective does not offer robustness benefits. To the best of our knowledge, this is the first study that focuses on group robustness in RLHF preference optimization; (ii) We analyze the theoretical aspects of GRPO by examining the convergence and the feasibility of finding optimal solutions within the log-linear policy class. (iii) We present a tailored algorithm to tackle this robust optimization challenge, providing convergence guarantees for certain loss functions; (iv) We show the versatility of our approach by demonstrating how our algorithm can be utilized with other reward-free preference optimization methods such as Identity Preference Optimization (IPO) [1]. In particular, for the GR-IPO objective optimized over the log-linear policy class, we derive a closed-form weighted regression update for the policy parameters rather than a gradient update. To the best of our knowledge, this is a novel contribution towards efficient fine-tuning through preferential data; (v) Our empirical evaluations across synthetic datasets, real-world data, and publicly available LLMs show that the proposed GRPO significantly improves performance for the worst-performing groups, reduces loss imbalances across groups, and increases probability accuracies compared to non-robust baselines.

## 2 Background

We address the challenge of fine tuning a large language model (LLM) to align with user preferences. This process usually follows the Reinforcement Learning from Human Feedback (RLHF) protocol, using either an explicit reward model ([3, 34, 45, 60]) or an implicit reward model ([37]). RLHF typically comprises three key phases: (i) supervised fine-tuning of an initial (pre-trained) large-language model, (ii) reward learning, and (iii) RL fine-tuning.

In the *supervised fine-tuning phase* (SFT), the goal is to fine-tune a pre-trained LLM on a specific high-quality dataset suited for the downstream task of interest. It results in a probabilistic model expressing the probability of the response $y$ given a prompt $x$ as $\pi_{\text{ref}}(y|x)$. Subsequently, in the *reward learning phase*, the goal is to learn a reward model from a dataset of prompts $x$ and responses $y_w, y_l$, with $y_l \prec y_w \mid x$ meaning that human labellers preferred $y_w$ over $y_l$. It is typically assumed that preferences follow some choice models with an *unknown* reward (utility) $r^*(x, y)$ function. A popular model is the Bradley-Terry model [5] that assumes the preference distribution $p$ admits the following form :

$$p(y_1 \prec y_2 \mid x) = \frac{\exp(r^*(x, y_2))}{\exp(r^*(x, y_1)) + \exp(r^*(x, y_2))}. \tag{1}$$

Based on the above model, a maximum likelihood estimate of the reward function is obtained as:

$$\min_r \{\mathcal{L}_R(r; \mathcal{D}) \coloneqq -\mathbb{E}_{(x, y_w, y_l) \sim \mathcal{D}} \left[\log\left(\sigma\left(r(x, y_w) - r(x, y_l)\right)\right)\right]\}, \tag{2}$$

where $\sigma(\cdot)$ is the sigmoid function and $\mathcal{D}$ represents the dataset consisting of $\{(x, y_w, y_l)\}$.

Then, in the RL fine-tuning phase the objective is to train a policy $\pi$ that maximizes the learned reward function. Simultaneously, the policy should stay closely aligned with the reference, $\pi_{\text{ref}}$, as

quantified by the KL divergence, leading to the following KL-regularized optimization problem:

$$\max_{\pi} \ \mathbb{E}_{x \sim \mathcal{P}_x} \Big[ \mathbb{E}_{y \sim \pi} \left[ r(x, y) \right] - \beta \mathrm{KL} \left[ \pi(y|x) \| \pi_{\mathrm{ref}}(y|x) \right] \Big]. \tag{3}$$

**Direct Preference Optimization (DPO).** The recent approach proposed by [37] exploits the closed-form solution of the problem in Equation (3) and sidesteps the explicit modelling of rewards to directly optimize the policy. Specifically, under the Bradley-Terry preference model, the reward function can be expressed directly in terms of the optimal policy $\pi^*$ as follows:

$$r(x, y) = \beta \log \frac{\pi^*(y \mid x)}{\pi_{\mathrm{ref}}(y \mid x)} + \beta \log Z(x), \tag{4}$$

for a partition function $Z(x) = \sum_y \pi_{\mathrm{ref}}(y|x) \exp(\frac{1}{\beta} r(x, y))$. Via a change of variable, finding the optimal reward function in Equation (2) is equivalent to finding the optimal policy $\pi^*$ utilizing the given set of preference data $\mathcal{D}$. With a slight abuse of notation, we use $\pi$ to denote $\pi^*$. Denote $h_\pi(x, y_w, y_l) := \log(\frac{\pi(y_w|x)}{\pi_{\mathrm{ref}}(y_w|x)}) - \log(\frac{\pi(y_l|x)}{\pi_{\mathrm{ref}}(y_l|x)})$. Then, Equation (2) translates into the DPO loss:

$$\mathcal{L}_{\mathrm{DPO}}(\pi, \mathcal{D}) = - \mathbb{E}_{(x, y_w, y_l) \sim \mathcal{D}} \Big[ \log \big( \sigma(\beta \cdot h_\pi(x, y_w, y_l)) \big) \Big]. \tag{5}$$

With a parameterized policy model $\pi_\theta$, minimizing the DPO loss involves calculating the gradient over $\theta$ using backpropagation and the log-probabilities of each completion, $y_w$ and $y_l$, given the prompt $x$ for both the policy $\pi_\theta$ and the reference policy $\pi_{\mathrm{ref}}$.

# 3 Group Robust Preference Optimization

In this section, we discuss Group Robust Preference Optimization (GRPO), i.e., instead of learning a reward function that maximizes the likelihood, we aim to derive (implicitly) a robust reward function and subsequently learn a robust DPO policy.

**Group Preferences.** Suppose that preferences come from an underlying latent reward $r^*(x, y, g)$, with $g \in \mathcal{G} = \{1, 2, \ldots, K\}$ indexing the groups. When group information is available (e.g., as a text), we can represent the reward as $r^*(x_g, y)$, where $x_g = x \oplus g$ denotes merging[1] of the prompt with group information (e.g., string concatenation). We continue to apply a Bradley-Terry model as described in Equation (1), substituting $x$ with $x_g$. Moreover, we assume access to a collective dataset $\mathcal{D} = \bigcup_{g=1}^{K} \mathcal{D}_g$ where $\mathcal{D}_g = \{(x_g^{(i)}, y_w^{(i)}, y_l^{(i)})\}_{i=1}^{N_g}$ with the available group information. Additionally, our dataset accommodates the exposure of different groups to identical prompts, meaning that the same $x$ can appear across various groups $g$ in our dataset, and these groups may favor different responses $y$.

Given such $\mathcal{D}$, although one may obtain a common reward model using Equation (2), it could result in poor generalization for particular groups, especially with significant group-wise disparities in the data (see Figure 1). Such disparities might stem from imbalanced data across groups or difficulties associated with learning different groups.

**GRPO Objective.** Consequently, we propose to measure the alignment of the reward model on the *worst-case group* loss:

$$\max_{g \in G} \ \mathcal{L}_R(r; \mathcal{D}_g). \tag{6}$$

Incorporating the reward expression from (Equation (4)) into (Equation (6)), we establish the *group robust preference optimization* (GRPO) objective for a specified policy $\pi$:

$$\mathcal{L}_{\mathrm{GR}}(\pi) := \max_{g \in \mathcal{G}} \mathcal{L}_{\mathrm{DPO}}(\pi, \mathcal{D}_g) = \max_{g \in \mathcal{G}} \Big( - \mathbb{E}_{(x_g, y_w, y_l) \sim \mathcal{D}_g} \Big[ \log \big( \sigma(\beta h_\pi(x_g, y_w, y_l)) \big) \Big] \Big). \tag{7}$$

Leveraging the equivalent formulation of maximizing over discrete set, the GRPO problem becomes

$$\min_{\pi} \mathcal{L}_{\mathrm{GR}}(\pi) = \min_{\pi} \max_{\alpha \in \Delta_{K-1}} \sum_{g=1}^{K} \alpha_g \Big( - \mathbb{E}_{(x_g, y_w, y_l) \sim \mathcal{D}_g} \Big[ \log \big( \sigma(\beta h_\pi(x_g, y_w, y_l)) \big) \Big] \Big), \tag{8}$$

---

[1]The group information added to the prompt can represent, e.g., an index or a textual description of the group.

where $\Delta_{K-1}$ represents the $(K-1)$-dimensional simplex of probabilities.[2] The inner maximization becomes a linear programming over simplex such that $\alpha$ represents the *weights* of groups. In addition, it forms a two-player zero-sum game (see Section 3.1), where the policy $\pi$ and $\alpha$ act as opponents with inversely related payoffs. The DPO loss (logistic log loss) in Equation (7) can be replaced with alternatives like hinge or squared loss (see [47]). We label this objective GR-DPO when using DPO loss, and explore GRPO with squared loss in Section 4.1.

**Applications.** In this study, we do not assume any specific distribution for groups $\mathcal{D}_g$. The collection of prompts per group, $\mathcal{P}_{x_g}$, may have varying degrees of overlap. The GRPO framework accommodates both distinct and overlapping prompt scenarios across different groups.

Apart from human groups, GRPO can be useful in scenarios where groups $\mathcal{D}_g$ represent distinct tasks or topics within preference datasets, like helpful/harmful, truthful/unanswerable instances, or domain-specific categories (e.g., math, physics, chemistry). Typically, these prompt distributions $\{\mathcal{P}_{x_g}\}_{g=1}^N$ are disjoint, and GRPO seeks to optimize performance even across the most challenging categories.

GRPO is also applicable in scenarios where groups reflect diverse user preferences for a *shared* set of prompts, with the goal of achieving equitable performance across user groups. This contrasts with non-robust DPO, which aims to optimize preferences on average and might overlook minority groups.

Lastly, we acknowledge that the max-min objective of Equation (8) might be overly conservative, potentially degrading average performance. We explore a more balanced approach between worst-case and standard preference optimization objective in Appendix B.4.

## 3.1 Further Discussion and Insights

This section provides two insights regarding the GR-DPO loss in Equation (8).

**Log-linear policy class.** The zero-sum game perspective allows us to explore the presence of a Nash equilibrium, serving as a benchmark for convergence during the policy optimization process. Given that the domain of $\alpha$ is a simplex $\Delta_K$ (in Equation (8)), we further define a parameterized policy class $\Pi_\theta$ for the policy $\pi_\theta$. We assume that the parameterized policy $\pi_\theta$ is of the form $\pi_\theta(y \mid x) = \frac{\exp f_\theta(x,y)}{\sum_{y \in \mathcal{Y}} \exp f_\theta(x,y)}$, where $f_\theta$ is a linear function or a neural network, and $\theta$ belongs to a convex set $\Theta$.

In LLM fine-tuning, sometimes practitioners concentrate on modifying solely the final layer. It corresponds to a linear function, $f_\theta(x,y) = \theta^T \phi(x,y)$, with $\phi(x,y)$ denoting the embedding derived from the language model removing its last layer, and $\theta$ as the parameters of the last layer. When applying this linear parameterization, in conjunction with a uniform reference policy $\pi_{\text{ref}}$, the robust objective outlined in Equation (8) is as follows (details in Appendix B.1):

$$\min_{\theta \in \Theta} \max_{\alpha \in \Delta_{K-1}} \sum_{g=1}^K \alpha_g \left( -\mathbb{E}_{(x_g, y_w, y_l) \sim \mathcal{D}_g} \left[ \log \left( \sigma \left( \beta \langle \phi(x, y_w) - \phi(x, y_l), \theta \rangle \right) \right) \right] \right). \tag{9}$$

The objective defined in Equation (9) is concave with respect to $\alpha$ and convex with respect to $\theta$. This structure allows the invocation of the minimax theorem for convex-concave functions ([42]) to assert the existence of a Nash equilibrium.

**Proposition 3.1.** *Under log-linear parameterization of the policy class, there exists a Nash equilibrium for the group robust direct preference optimization problem in Equation* (9).

**Robust policy optimization.** The earlier derivation for the GR-DPO objective $\mathcal{L}_{\text{GR-DPO}}(\pi)$ relies on incorporating robustness in the reward modeling step (in Equation (7)) while using the solution to the non-robust KL-regularized reward maximization objective in Equation (4).

Interestingly, we can obtain the identical expression for $\mathcal{L}_{\text{GR-DPO}}(\pi)$ if incorporating robustness in the KL-regularized reward maximization objective and using the reward function learnt in a non-robust way. Consider the robust KL-regularized reward maximization

$$\max_{\pi} \min_{g \in \mathcal{G}} \mathbb{E}_{x_g \sim \mathcal{P}_{x_g}, y \sim \pi(\cdot \mid x_g)} \left[ r(x_g, y) - \beta \text{KL} \left[ \pi(y \mid x_g) || \pi_{\text{ref}}(y \mid x_g) \right] \right]. \tag{10}$$

The following proposition characterizes such an invariant property.

---

[2]From a distributionally-robust viewpoint, this is equivalent to defining the uncertainty set as $\left\{ \sum_{g=1}^K \alpha_g \mathcal{D}_g : \alpha \in \Delta_K \right\}$, and minimizing the worst-case expected loss across the uncertainty set.

**Algorithm 1** Mirror Descent for Group Robust Preference Optimization (GRPO)

---
1: **Initialize:** Step size $\eta_\alpha$ for group weights $\alpha$, step size $\eta_\theta$ for policy $\pi$ with weights $\theta$, initial weights $\theta^{(0)}$ of the policy and weights over each group $\alpha^{(0)}$, Projection operator $\mathrm{P}_\Theta$
2: **Input:** Dataset $\mathcal{D}$ with size $N = |\mathcal{D}|$, group size $N_g$ for $g = \{1, 2, \cdots, K\}$, loss $l(\pi_\theta; \cdot)$
3: **for** $t = 1, \ldots, T$ **do**
4:     $\alpha' \leftarrow \alpha^{(t-1)}$
5:     $g \sim \mathrm{Categorical}(N_1/N, \cdots, N_K/N), (x_g, y_w, y_l) \sim \mathcal{D}_g$
6:     $\alpha'_g \leftarrow \alpha'_g \exp \eta_\alpha \left( \frac{N \cdot l(\pi_{\theta(t-1)}; (x_g, y_w, y_l))}{N_g} \right)$    // Update weights for group $g$
7:     $\alpha^{(t)} \leftarrow \alpha' / \sum_{g'} \alpha'_{g'}$    // Renormalize $\alpha$
8:     $\theta^{(t)} \leftarrow \mathrm{P}_\Theta \left( \theta^{(t-1)} - \eta_\theta \left( \frac{N \alpha_g^{(t)} \nabla_\theta l(\pi_{\theta(t-1)}; (x_g, y_w, y_l))}{N_g} \right) \right)$ // Use $\alpha$ to update $\theta$
9: **end for**
10: **Return:** Output the robust policy $\pi(\theta^{(T)})$

---

**Proposition 3.2.** *Substituting the closed-form solution of the robust KL-regularized policy maximization problem (Equation* (10)*) into the robust reward maximization objective in Equation* (6) *leads to the same group robust DPO loss* $\mathcal{L}_{\mathrm{GR-DPO}}$ *in Equation* (8) *.*

The analysis leverages the fact that the optimal policy of Equation (10) is identical to the solution of the non-robust KL-regularized reward maximization in Equation (4) and is derived in Appendix B.2.

## 4 Algorithm

In this section, we discuss the policy optimization algorithm for solving the group robust DPO problem in Equation (8). In particular, we aim to design an algorithm that performs updates in the parameterized space $\Theta \subset \mathbb{R}^d$, i.e., updating $\theta$ of the parameterized policy $\pi_\theta$. Leveraging the perspective of the 2-player zero-sum game, we propose an alternating updating algorithm wherein one updates $\alpha$ and $\theta$ alternatively. We summarize the overall approach in Algorithm 1, which we discuss and analyze next.

We employ the DPO loss $l(\pi_\theta; \cdot) = \log\left(\sigma(\beta h_{\pi_\theta}(\cdot))\right)$ (Equation (5)) in Algorithm 1, however, our algorithm can support other preference optimization losses (see Section 4.1). The algorithm performs a gradient descent type update on $\theta$ and a deterministic mirror ascent on $\alpha$ using a Bregman divergence with the distance generating function as the KL divergence. Since the $\alpha$ lies in a simplex and the objective is linear, the update of $\alpha$ becomes multiplicative weights update with renormalization to a simplex via softmax (see Nemirovski et al. [32] for details). Further, the weights $\alpha$ are determined by the cumulative losses $l(\pi_\theta; \cdot)$ accrued by each group, ensuring that groups with higher cumulative losses get higher weights. The size of the group $N_g$ appears as the empirical distribution $\mathcal{D}_g$ involves $N_g$. We call it alternating update as the updated $\alpha^t$ is used in the update from $\theta^{t-1}$ to $\theta^t$. In particular, the gradient descent type update on $\theta$ is weighted by $\alpha$ in order to orient the update towards groups with higher losses. The projection operator $\mathrm{P}_\Theta$ ensures the updated $\theta^t$ lies within $\Theta$.

**What does the weighted DPO update do?** In Line 9 in Algorithm 1, the algorithm performs parameter updates based on the weighted gradients. By using the DPO loss, i.e., $l(\pi_\theta; \cdot) = \log\left(\sigma(\beta h_{\pi_\theta}(\cdot)\right)$ (see Equation (5)), we obtain the following gradient update expression ignoring the $N/N_g$ constant

$$\alpha_g^{(t)} \nabla_\theta l(\pi_{\theta(t-1)}; (x_g, y_w, y_l)) = \alpha_g^{(t)} \nabla_\theta \log\left(\sigma(\beta h_{\pi_{\theta(t-1)}}(x_g, y_w, y_l))\right) \tag{11}$$

$$= \alpha_g^{(t)} \sigma\left(r_{\theta(t-1)}(x_g, y_l) - r_{\theta(t-1)}(x_g, y_w)\right) \times [\nabla_\theta \log \pi_{\theta(t-1)}(y_w|x_g) - \nabla_\theta \log \pi_{\theta(t-1)}(y_l|x_g)].$$

The final term plays the critical role of enhancing the likelihood of the preferred response while simultaneously diminishing the likelihood of the rejected response. This adjustment is proportional to the disparity in rewards between the two responses. Moreover, the inclusion of $\alpha_g$ is pivotal for ensuring group robustness. This coefficient scales the gradient w.r.t. $\theta$ based on the cumulative loss previously received by all samples within a specific group. Such a mechanism ensures that the model's focus is increasingly directed towards groups that have historically suffered higher losses. Additionally, the scaling factor $N_g$ guarantees that groups with a smaller volume of data do not face a disadvantage. We defer further details in obtaining the gradient update expression to Appendix B.3.

We demonstrate the global convergence with the following proposition.

**Proposition 4.1.** *Suppose that the loss $l(\cdot; (x_g, y, y'))$ is non-negative, convex, $B_\nabla-$Lipschitz continuous, and bounded by $B_l$ for all $(x_g, y, y') \in \mathcal{X} \oplus \mathcal{G} \times \mathcal{Y} \times \mathcal{Y}$ and $\|\theta\|_2 \leq B_\Theta$ for all $\theta \in \Theta$ with convex $\Theta \subset \mathbb{R}^d$. The error of the average iterate of Algorithm 1, i.e., $\pi_{\bar{\theta}(1:T)} = \frac{1}{T}\sum_{t=1}^{T}\theta^t$, satisfies*

$$\mathbb{E}[\mathcal{L}_{\mathrm{GR}}(\pi_{\bar{\theta}(1:T)})] - \min_{\theta \in \Theta}\mathcal{L}_{\mathrm{GR}}(\pi_\theta) = \mathcal{O}(T^{-1/2}).$$

We defer the proof of this proposition to Appendix E. The analysis follows from an adaptation of the analysis in Nemirovski et al. [32] for the proposed sampling strategy in Algorithm 1[3]. We note that when fine-tuning only the final layer of a LLM, the output policy exists within the log-linear policy class (see Section 3.1), and the corresponding loss function satisfies the assumptions in Proposition 4.1 (see Lemma E.1).

## 4.1 Group Robust Identity Preference Optimization

The standard regularized reward maximization objective (Equation (3)) in DPO [37], tends to overlook the KL-regularization and learn deterministic policies. This learned policy assigns preference probability one to winning responses in the data which is often not realistic (see [1][Section 4.2] and Appendix C). Recently, Azar et al. [1] show that the standard regularized reward maximization objective (Equation (3)) in DPO [37] tends to overlook the KL-regularization and learn deterministic policies (see [1, Section 4.2] and Appendix C). They thus propose an alternative approach called *Identity Preference Optimization* (IPO) that is more likely to learn a randomized policy which assigns appropriate probability to the preferred response and prevents overfitting. Following a similar derivation as we did for group robust DPO with details given in Appendix C, we develop the corresponding group robust IPO (GR-IPO):

$$\min_{\pi}\ \mathcal{L}_{\mathrm{GR}}(\pi) := \max_{g \in \mathcal{G}}\mathcal{L}_{\mathrm{IPO}}(\pi, \mathcal{D}_g) = \max_{\alpha \in \Delta_{K-1}}\sum_{g=1}^{K}\alpha_g\Big(\mathbb{E}_{(x_g, y_w, y_l) \sim \mathcal{D}_g}\Big[h_\pi(x_g, y_w, y_l) - \tfrac{1}{2\beta}\Big]^2\Big).$$

For the log-linear policy class (introduced in Section 3.1), the objective function simplifies to

$$\min_{\theta \in \Theta}\max_{\alpha \in \Delta_{K-1}}\sum_{g=1}^{K}\alpha_g\Big(\mathbb{E}_{(x_g, y_w, y_l) \sim \mathcal{D}_g}\Big[\big(\langle\phi(x_g, y_w) - \phi(x_g, y_l), \theta\rangle - \tfrac{1}{2\beta}\big)^2\Big]\Big).$$

To solve the GR-IPO above, it suffices to use Algorithm 1 with slight modifications, see Algorithm 2 in Appendix C. In particular, the update of $\theta$ is replaced by a weighted regression update:

$$\hat{\theta} \leftarrow \arg\min_{\theta \in \Theta}\sum_{(x_g, y_w, y_l) \sim \mathcal{D}}\Big[\frac{\alpha_g}{N_g}\big(\langle\phi(x_g, y_w) - \phi(x_g, y_l), \theta\rangle - \frac{1}{2\beta}\big)^2\Big].$$

For fixed $\alpha$, we show (in Appendix C) that such an update admits a closed-form solution:

$$\hat{\theta} = \frac{1}{2\beta}(S^T W S)^{-1}S^T W \mathbf{1} \quad \text{with} \quad W := \mathrm{Diag}\Big[\frac{\alpha_{g^{(1)}}}{N_{g^{(1)}}}, \ldots, \frac{\alpha_{g^{(N)}}}{N_{g^{(N)}}}\Big],$$

where $g^{(i)}$ is the group of each sample $i$, $N_{g^{(i)}}$ is the number of samples in group $g^{(i)}$ and $\mathbf{1}$ is a column vector of ones of dimension $N$. Here $S$ is a matrix $S := \Big[(\phi(x_g^{(1)}, y_w^{(1)}) - \phi(x_g^{(1)}, y_l^{(1)}))^T, \ldots, (\phi(x_g^{(N)}, y_w^{(N)}) - \phi(x_g^{(N)}, y_l^{(N)}))^T\Big]$. Each row of $S$ represents the difference in feature mappings $\phi$ of the preferred and less preferred response for each prompt. The group robust IPO (GR-IPO) algorithm is presented in Appendix C, and its empirical results are shown in Section 5.

## 5 Experiments

In this section, we study the empirical performance of our proposed Algorithm 1 on synthetic and real-world datasets[4]. First, we simulate multi-group data disparities by varying the size and

---

[3]We study different sampling strategies and their error rates in Appendix E and we present the most numerically stable one in Algorithm 1.

[4]Codes for synthetic and real-world experiments can be found at https://github.com/rsshyam/GRPO-bandits and https://github.com/rsshyam/GRPO, respectively.

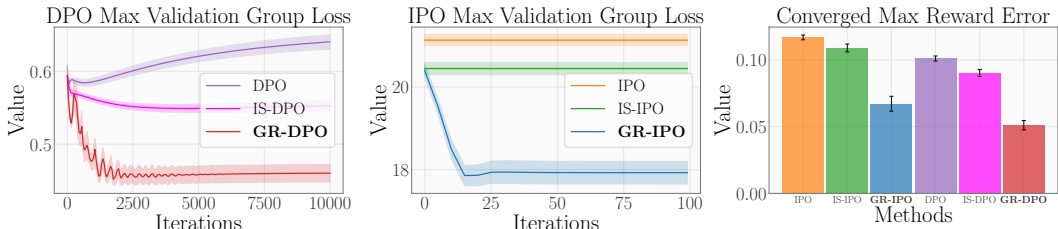

Figure 2: Synthetic experiments: Algorithm 1 (GR-DPO and GR-IPO) leads to a significantly lower worst-case validation loss and reward error compared to importance sampling (IS-DPO/IPO) and vanilla methods (DPO, IPO). Results refer to the scenario in which groups have different sizes and responses' distribution.

preference distributions of two synthetic groups. In the real-world setup, we study the alignment of an LLM to the real preferences of people from various countries. We examine whether GRPO aligns the LLM in a more equitable manner to reduce discrepancies in alignment among various groups. Finally, we demonstrate that performance is improved by explicitly addressing the grouped nature of the data during alignment.

## 5.1 Synthetic Experiments

We evaluate the performance of Algorithm 1 using synthetically generated group preference data for the loss function $l(\pi_{\theta_{:}})$ – either DPO loss or IPO loss and denote them as GR-DPO and GR-IPO, respectively. We compare them against vanilla DPO and IPO ([37, 1]), and the importance-sampling (IS) variants of DPO and IPO (where the loss of each datapoint is inversely weighted by its group data size).

**Experimental Setup.** Our experiments are designed to analyze settings where there exist multiple groups with distinct characteristics. We adapt the standard (non-group based) experimental setup proposed by [26] for the group preferences setting by incorporating group information into the reward function $r : \mathcal{X} \times \mathcal{Y} \times \mathcal{G} \to \mathbb{R}$. Here, $\mathcal{X}$ represents a two-dimensional state space $[0, 1] \times [0, 1]$, $\mathcal{Y}$ denotes a discrete action space $\{0, 1, 2, 3, \ldots, n\}$, and $\mathcal{G}$ signifies a discrete group space $\{0, 1, 2, \ldots, K\}$. The reward function, defined by the group-dependent feature vector $\phi(x, y, g)$ and parameter vector $\theta_g$, is given as $r(x, y, g) := \langle \phi(x, y, g), \theta_g \rangle$, while the feature vectors $\phi(x, y, g)$ have a coordinate-flipped relationship and are defined in Appendix D.1.

We consider the following scenarios: **(i)** Groups are imbalanced in terms of size but have the same distribution over responses, **(ii)** Groups are balanced in terms of size but have different response distributions, and **(iii)** Groups are imbalanced in terms of size and also have different response distributions. Note that having different response distributions leads to a difference in the difficulty of learning, since groups with responses distant from each other (in terms of rewards or preference probabilities) are typically more distinguishable and easier to learn. We discuss in Appendix D.2 how we generate these three scenarios.

**Implementation.** Leveraging the linearity in the reward model, we utilize a log-linear policy class parameterized by $\theta$: $\pi_\theta(y|x) = \frac{\exp \langle \phi(x,y,g), \theta \rangle}{\sum_{y' \in \mathcal{Y}} \exp \langle \phi(x,y',g), \theta \rangle}$. We run Algorithm 1 for both DPO and IPO loss relative to the policy class detailed above with a dataset of 300 action pairs with preferences.

**Evaluation Metrics.** We use the following criteria to assess the performance of the algorithms:

*Max Validation Loss.* For each group $g$, with preference data denoted as $(x^i, y_w^i, y_l^i, g)_{i=1}^{N_g}$, where $N_g$ is the number of data points in the group, we compute the DPO/IPO validation loss separately for each group and identify the maximum loss among them in each run.

*Max Reward Error.* This metric compares the true reward of the optimal action determined by $\theta_g^*$ with that of the action deemed optimal by estimate $\hat{\theta}$ for each group, and identifies the maximum error across all groups in each run. In particular, for data in the form $(x^i, g)_{i=1}^{N_g}$, which includes only states and groups, we calculate reward errors for every group $g$ as follows: $\mathbb{E}_{(x,g) \sim (x^i, g)_{i=1}^{N_g}} [\max_y \langle \phi(x, y, g), \theta_g^* \rangle - \langle \phi(x, \arg\max_y \langle \phi(x, y, g), \hat{\theta} \rangle, g), \theta_g^* \rangle]$.

**Results.** We present the average performance of Algorithm 1 (error bars over 20 seeds) alongside baseline methods in Figure 2 for scenario **(iii)**, while scenarios **(i)** and **(ii)** are Figures 4 and 5

in Appendix D.2 due to space constraints. Our findings indicate that the robust methods consistently surpass both vanilla and importance-sampling approaches. Notably, the robust methods demonstrate significant superiority in uneven group scenarios, where the importance-sampling technique falls short as it exclusively deals with data imbalance.

## 5.2   Global Opinion Experiments

For the real-data experiments, we consider the survey dataset *GlobalOpinionQA* ([15]) and the publicly available Gemma-2B model [48]. [5] The data contains multiple choice questions answered by participants from various countries, amounting to 2,554 questions covering various topics, including politics, media, technology, religion, race, and ethnicity. For each question, the dataset provides a probability vector over the choices, signifying the percentage of people from a particular country choosing each option. Note that this probability vector would be different for different countries. Hence, the goal is to align the LLM to the probability vector corresponding to each country in a robust manner.

We consider the following five countries in the dataset: Nigeria, Egypt, India, China and Japan, with data sizes 572, 570, 376, 309, and 712, respectively. We construct our training set as follows: For the SFT training, we choose the best option (the choice with the highest probability) as the target. For both IPO and GR-IPO training, we consider the best option as the winning response and another randomly chosen option as the losing response. We outline the exact prompt we use in Appendix D.

We run the SFT training for one epoch over the training data on the pre-trained Gemma-2B model. For both IPO/GR-IPO training we use the AdamW [27] optimizer with adaptive learning rates. For SFT/IPO/GR-IPO training, we apply the LoRA strategy to fine-tune all layers of the model. We then evaluate both the methods based on the worst group loss and accuracy. Here, the loss refers to the IPO loss for each group and the accuracy refers to the percentage of winning response and losing response pairs correctly ordered by the learned preference function (Equation (35)). We defer further training and hyperparameter details to Appendix D.

**Results.** We present the average performance of GR-IPO over five seeds alongside IPO in Figure 3 (top plots). Our findings indicate that GR-IPO outperforms IPO in terms of maximum group loss and minimum group reward accuracies. Moreover, GR-IPO effectively reduces the imbalance in loss values among different groups. Additionally, we observe an improvement in log-probability accuracies (which measure if the probability assigned by the fine-tuned model is higher for the winning response compared to the losing response) for both IPO and GR-IPO, with GR-IPO demonstrating better alignment for the worst-performing group compared to IPO.

**Insights.** We further note that the worst-performing groups are Groups-2,5, as shown in Figure 3. GR-IPO improves the loss for these groups by assigning more weight to them, as illustrated in Figure 3 (bottom middle plot). Additionally, we plot the initial log-probability accuracies for different groups in Figure 3 (bottom right plot), assessing how accurately the SFT model classifies the winning versus losing response for different groups. It is evident that Groups-2,5 are already underperforming. Given that SFT training converges within one epoch without any discrepancies between groups, this indicates that the base LLM inherently struggles with classifying responses for Groups-2,5. However, by employing GR-IPO, we have mitigated the imbalance in the performance of the fine-tuned model.

## 6   Conclusions

We formalize the problem of robustly aligning an LLM to preference distributions from diverse groups. To tackle the same, we introduced GRPO, a group robust formulation of reward-free RLHF, aiming to minimize worst-case loss among groups. We explored the theoretical aspects of GRPO and demonstrated its improved robust alignment performance through various experiments. We believe our approach will be highly valuable for future tailored LLM fine-tuning, specifically aimed at aligning with the needs of diverse teams and user groups. In a broader context, it holds promise for mitigating biases and discrepancies across various societal groups encountered in the task-specific adaptation of LLMs.

**Limitations.** When the dataset is balanced among groups and difficulty levels are comparable, our GRPO approach does not offer a significant advantage over standard reward-free RLHF algorithms. In addition, minimax methods often improve the worst group's performance at the cost of reducing the average or best group's performance. Our proposed GRPO formulation (see Equation (8)),

---

[5]The model and dataset are available on Hugging Face at https://huggingface.co/google/gemma-2b and https://huggingface.co/datasets/Anthropic/llm_global_opinions, respectively.

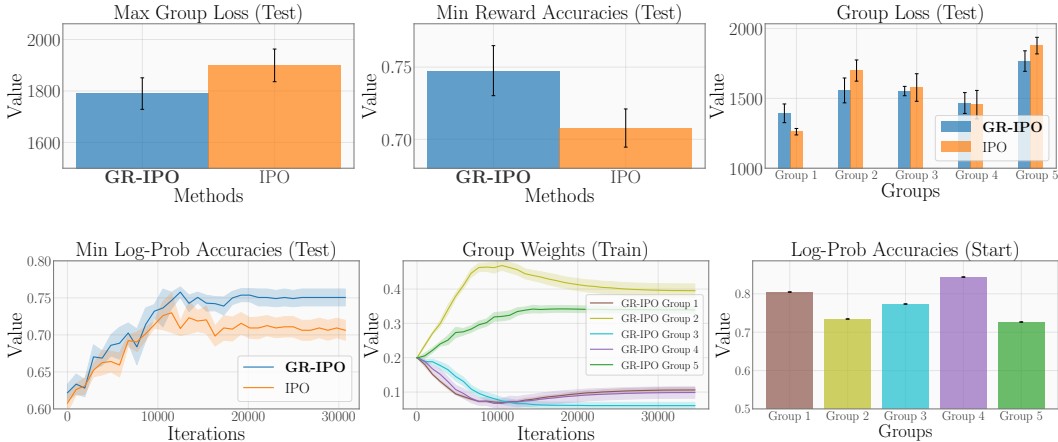

Figure 3: Global opinion data: **Top plots:** GR-IPO leads to better worst-case final test loss and reward accuracy compared to IPO. Moreover, it leads to more balanced losses across the different groups, reducing the gap between best and worst-group loss (Group-1 vs. Group-5). **Bottom plots:** Log-prob. accuracy (left plot) and group weights (middle plot) during GR-IPO training. GR-IPO increases the weight on worse-performing groups (Groups-2,5) and decreases it on high-performing ones (Groups-1,3,4), leading to better worst-case accuracy. Groups-2,5 are the ones with worse log-prob. accuracy at the beginning of training (right plot with a random subset of the training data). We show the corresponding end-of-training log-prob. accuracies for GR-IPO in Figure 13 of Appendix D.

stemming from a minimax framework, will comply with the same property. Hence, in scenarios where optimizing worst-case performance is less critical, we define a *trade-off parameter* to balance between the worst-case performance and the average performance. This modified objective and the necessary algorithmic changes are elaborated in Appendix B.4. The appropriate tuning of the trade-off parameter for the specific application remains a subject for future investigation. Further, we focus on settings with known groups, which are common in pluralistic alignment datasets and tasks (see [57, 44]). When groups are unknown, one can still undertake several approaches such as clustering, representation learning, feature analysis, expert consultations, etc., to help uncover group structures in the data.

# 7   Acknowledgments

PGS was gratefully supported by ELSA (European Lighthouse on Secure and Safe AI) funded by the European Union under grant agreement No. 101070617. YH was supported as a part of NCCR Automation, a National Centre of Competence (or Excellence) in Research, funded by the Swiss National Science Foundation (grant number 51NF40_225155). IB was supported by the EPSRC New Investigator Award EP/X03917X/1; the Engineering and Physical Sciences Research Council EP/S021566/1; and Google Research Scholar award. SSR was supported by Department of Electronic and Electrical Engineering and the Institute of Communications and Connected Systems at UCL. The authors would like to thank William Bankes, Seongho Son, Matthieu Zimmer, Afroditi Papadaki, Eduardo Pignatelli, and Nagham Osman for the useful discussion.

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

# Appendix

The supplementary section constitutes the following: An extended related work section in Appendix A, proofs for the theoretical insights and the weighted DPO gradient expression in Appendix B, an overview of IPO and the simplification for robust IPO in Appendix C, detailed experimental setup in Appendix D, and convergence proof for Algorithm 1 and other sampling strategies in Appendix E.

## A   Additional Related Work

We detail a more extensive related work exposition in this section.

The Reinforcement Learning from Human Feedback (RLHF) is a concrete way to adapt a LLM towards specific tasks or objectives using human feedback. This framework for fine-tuning LLMs is inspired from [11] that uses human feedback to improve reinforcement learning policies. For LLMs, it was first established in [60] and further developed in [45, 34]. It involves learning a reward function using the pairwise human feedback data that provides comparisons between responses for a given prompt. This reward learning is performed through the Bradley-Terry model [5]. This learned reward model is then used as an objective for optimizing the LLM policy using Proximal Policy Optimization [40].

However, obtaining high-quality human feedback data is expensive and several works such as [35, 2, 25] study the usage of AI feedback to reduce reliance on human input. Further, studies like [13, 28, 22] analyze the usage of active learning strategies to minimize the amount of human feedback data required to perform RLHF. For a comprehensive overview and perspective of the RLHF topic, with detailed discussions on various approaches and gaps, we refer the reader to [24, 7, 23].

Due to the inherent issues of tuning the hyperparameters of a PPO ([50, 37]) and susceptible nature of reward models ([17, 49]), alternative approaches to the PPO-based RLHF have been proposed. Foremost of them is the rejection sampling fine-tuning ([14, 18, 49]) which is inspired by the best-of-n inference technique ([30]). The technique involves sampling $n$ responses per prompt and fine-tuning the model based on the highest scoring ones in terms of a learned reward function. Further works such as [20, 54] avoid reward learning and propose conditional Supervised Fine-Tuning (SFT) inspired by the reward-conditioned RL ([9]).

Another strategy proposed as an alternative to PPO-based RLPHF is DPO ( Rafailov et al. [37]), wherein one optimizes the policy directly based on human preferences, bypassing the need for learning a separate reward model. This method simplifies the training process and potentially reduces overfitting and model misalignment issues. Theoretical advancements in understanding the dynamics of DPO have been provided by [21, 53], who analyzed the convergence and stability of DPO policy learning. Some other works such as [1, 58, 47, 43, 16] also study a similar reward-free RLHF setup. In particular, [1] analyze the potential overfitting issues in DPO. Further, [16, 6] propose strategies that bypass the need for preference datasets. [38, 55, 29] adopt a game-theory perspective to learn Nash equilibrium policies. Typically, the reward-free RLHF approaches utilize a reference policy trained via supervised fine-tuning, which is then further optimized. However, [19] propose Monolithic Preference Optimization without reference model, bypassing supervised fine-tuning entirely. [59] address the RLHF problem from an MDP framework, introducing a DPO-type algorithm integrated with Proximal Policy Optimization (PPO). Additionally, works such as [51, 46, 10, 41, 56] investigate self-play preference optimization, where new data generated in each round by the current policy is used to train an improved policy. Our work utilizes a reward-free framework similar to [37, 1] but differs from all the above works in the following sense. Unlike previous works that assume a single preference distribution, we consider fine-tuning a LLM given data from multiple preference distributions across diverse groups. Further, we aim to fine-tune the LLM in a robust manner such that there is minimal imbalance in performance across all groups.

Robust language modeling technique in the context of language model was first studied in [33] to optimize performance over a wide-range of topics. It was further improved by [52] wherein they use smaller model to learn the weights/importance that needs to be assigned to different topics and train a larger model based on these weights. Both works aim to minimize the worst group loss based on the group-DRO approach and focus on the pre-training aspects of language models. Concrete theoretical study of this group-DRO approach was performed in [39] which studies the minimax formulation of this group-DRO problem based on [31]. In the context of online batch selection, [4] propose a robust

data downsampling method that performs class-aware priority reweighting to reduce training costs while preserving worst-class performance.

In the RLHF setup, recent research has increasingly focused on enhancing the robustness of policy learning to address certain inherent gaps in the RLHF framework as detailed in [7] such as reward hacking and model misspecification. In particular, [3] formulated a weighted group loss for reward model learning w.r.t. a parameter $\lambda$ that determines the importance assigned to a group and apply it to the anthropic Harmless vs Helpful data ([6]). In a similar manner,[12] design a weighted objective for policy optimization w.r.t. parameter $\lambda$ to trade-off the importance between task reward maximization and safety cost minimzation. Another work [8] consider alignment to different sub-populations by learning multiple reward functions corresponding to each sub-population and learning a robust policy (max-min) w.r.t. the reward functions. Distinct from previous works, our approach circumvents reward model learning and integrates group robustness directly into the reward-free tuning framework. This allows us to directly learn a robust policy without the need for learning reward models. To learn this robust policy, we present a concrete algorithm that adaptively assigns weights to the losses of different groups and optimizes the policy to minimize this weighted loss. Additionally, our algorithm features a novel gradient estimator specifically designed for the group robust DPO problem.

Other studies addressing robustness in preference optimization, such as [21] and [26], focus on different facets of robustness, including robustness to noise and resilience against out-of-preference data. In the non-robust group preference alignment setup, [57] explores learning group preferences with LLMs by incorporating group information into prompts. However, their approach does not involve fine-tuning a LLM but training a separate transformer module that optimally selects an example sequence of prompts, LLM responses, and group preferences for in-context learning using a LLM. Also, Wang et al. [49] consider alignment with user preferences assuming that each user/group has varied importance over the distinct metrics in their multi-objective reward model. The output policy is trained to output a response based on both the prompt and the importance/weights over the individual metrics. In contrast, the distinctive feature of our method is that we are not modeling multi-reward objectives but consider a reward-free setting. Specifically, we consider the robust alignment problem optimizing for the worst group performance. And, our methodology directly models each group's preferences through a group-dependent latent reward model where the group dependency is injected through the prompt.

## B    Theoretical Insights

In this section, we detail the proofs for the theoretical insights elucidated in Section 3 and the weighted DPO gradient expression discussed in Section 4.

### B.1    Robust Objective for the Log-linear Policy Class

Here, we describe obtaining the robust objective for the log-linear policy class as in Equation (9). Starting from the robust objective for a general policy class in Equation (8), $\mathcal{L}_{\text{GR}}$ can be specialized as follows:

$$
\begin{aligned}
\mathcal{L}_{\text{GR}}(\pi) &= \max_{\alpha \in \Delta_{K-1}} \sum_{g=1}^{K} \alpha_g \Big( -\mathbb{E}_{(x_g, y_w, y_l) \sim \mathcal{D}_g} \Big[ \log \Big( \sigma(\beta h_\pi(x_g, y_w, y_l)) \Big] \Big) \\
&\overset{(i)}{=} \max_{\alpha \in \Delta_{K-1}} \sum_{g=1}^{K} \alpha_g \Big( -\mathbb{E}_{(x_g, y_w, y_l) \sim \mathcal{D}_g} \Big[ \log \Big( \sigma(\beta \log(\tfrac{\pi(y_w|x_g)}{\pi_{\text{ref}}(y_w|x_g)}) - \beta \log(\tfrac{\pi(y_l|x_g)}{\pi_{\text{ref}}(y_l|x_g)})) \Big) \Big] \Big) \\
&\overset{(ii)}{=} \max_{\alpha \in \Delta_{K-1}} \sum_{g=1}^{K} \alpha_g \Big( -\mathbb{E}_{(x_g, y_w, y_l) \sim \mathcal{D}_g} \Big[ \log \Big( \sigma(\beta \log(\pi(y_w|x_g)) - \beta \log(\pi(y_l|x_g))) \Big) \Big] \Big)
\end{aligned}
$$

---

[6]The data is publicly avaiable in Huggingface website https://huggingface.co/datasets/Anthropic/hh-rlhf

$$\overset{(iii)}{=} \max_{\alpha \in \Delta_{K-1}} \sum_{g=1}^{K} \alpha_g \Big( - \mathbb{E}_{(x_g, y_w, y_l) \sim \mathcal{D}_g} \Big[ \log \Big( \sigma(\beta \log(\frac{\exp f_\theta(x_g, y_w)}{\sum_{y \in \mathcal{Y}} \exp f_\theta(x_g, y)}) -$$

$$\beta \log(\frac{\exp f_\theta(x_g, y_l)}{\sum_{y \in \mathcal{Y}} \exp f_\theta(x_g, y)}) ) \Big] \Big)$$

$$\overset{(iv)}{=} \max_{\alpha \in \Delta_{K-1}} \sum_{g=1}^{K} \alpha_g \Big( - \mathbb{E}_{(x_g, y_w, y_l) \sim \mathcal{D}_g} \Big[ \log \Big( \sigma(\beta \log(\exp \theta^T \phi(x_g, y_w)) -$$

$$\beta \log(\exp \theta^T \phi(x_g, y_l))) \Big] \Big)$$

$$= \max_{\alpha \in \Delta_{K-1}} \sum_{g=1}^{K} \alpha_g \Big( - \mathbb{E}_{(x_g, y_w, y_l) \sim \mathcal{D}_g} \Big[ \log \Big( \sigma\big(\beta \langle \phi(x_g, y_w) - \phi(x_g, y_l), \theta \rangle\big) \Big) \Big] \Big).$$

Here, (i) follows from the definition of $h_\pi(x_g, y_w, y_l) = \log(\frac{\pi(y_w|x_g)}{\pi_{\text{ref}}(y_w|x_g)}) - \log(\frac{\pi(y_l|x_g)}{\pi_{\text{ref}}(y_l|x_g)})$, (ii) follows from the fact that for a uniform reference policy $\pi_{\text{ref}}(y_w|x_g) = \pi_{\text{ref}}(y_l|x_g)$, (iii) follows from substituting $\pi_\theta(y|x_g) = \frac{\exp f_\theta(x_g, y)}{\sum_{y \in \mathcal{Y}} \exp f_\theta(x_g, y)}$, and (iv) follows from substituting $f_\theta(x_g, y) = \theta^T \phi(x_g, y)$.

## B.2 Robust KL-Regularized Policy Maximization

In this section, we consider the robust version of the classical KL-regularized reward maximization objective (Equation (3)) detailed in Equation (10) for a reward function $r(x_g, y)$, reference policy $\pi_{\text{ref}}$, and a general class of policies $\Pi$.

$$\max_{\pi \in \Pi} \min_{g \in \mathcal{G}} \mathbb{E}_{x_g \sim \mathcal{P}_{x_g}, y \sim \pi(\cdot|x_g)} \big[ r(x_g, y) \big] - \beta \text{KL}[\pi(y \mid x_g) || \pi_{\text{ref}}(y \mid x_g)]. \tag{12}$$

**Lemma B.1.** *The optimal policy $\pi^*$ for the robust KL-regularized reward maximization objective (Equation (12)) for a reward function $r(x_g, y)$, reference policy $\pi_{ref}$ is*

$$\pi^*(y|x_g) = \frac{1}{Z(x_g)} \pi_{ref}(y|x_g) \exp\Big(\frac{r(x_g, y)}{\beta}\Big), \tag{13}$$

*where $Z(x_g) = \sum_y \pi_{ref}(y|x_g) \exp(\frac{1}{\beta} r(x_g, y))$ is a partition function.*

*Proof.* We recast Equation (12) based on Equation (8) and perform the following analysis similar to [37][Appendix A.1]:

$$\max_{\pi \in \Pi} \min_{\alpha \in \Delta_{K-1}} \sum_{g=1}^{K} \alpha_g \Big( \mathbb{E}_{x_g \sim \mathcal{P}_{x_g}} \Big[ \mathbb{E}_{y \sim \pi(\cdot|x_g)} \big[ r(x_g, y) \big] - \beta D_{KL} \big[ \pi(y|x_g) || \pi_{\text{ref}}(y|x_g) \big] \Big] \Big) \tag{14}$$

$$= \max_{\pi \in \Pi} \min_{\alpha \in \Delta_{K-1}} \sum_{g=1}^{K} \alpha_g \Big( \mathbb{E}_{x_g \sim \mathcal{P}_{x_g}} \mathbb{E}_{y \sim \pi(\cdot|x_g)} \big[ r(x_g, y) - \beta \log\big(\frac{\pi(y|x_g)}{\pi_{\text{ref}}(y|x_g)}\big) \big] \Big) \tag{15}$$

$$= -\beta \min_{\pi \in \Pi} \max_{\alpha \in \Delta_{K-1}} \sum_{g=1}^{K} \alpha_g \Big( \mathbb{E}_{x_g \sim \mathcal{P}_{x_g}} \mathbb{E}_{y \sim \pi(\cdot|x_g)} \big[ \log\big(\frac{\pi(y|x_g)}{\pi_{\text{ref}}(y|x_g)}\big) - \frac{r(x_g, y)}{\beta} \big] \Big) \tag{16}$$

$$= -\beta \min_{\pi \in \Pi} \max_{\alpha \in \Delta_{K-1}} \sum_{g=1}^{K} \alpha_g \Big( \mathbb{E}_{x_g \sim \mathcal{P}_{x_g}} \mathbb{E}_{y \sim \pi(\cdot|x_g)} \big[ -\log(Z(x_g)) + \log\big(\frac{\pi(y|x_g)}{\frac{1}{Z(x_g)} \pi_{\text{ref}}(y|x_g) \exp(\frac{r(x_g, y)}{\beta})}\big) \big] \Big). \tag{17}$$

Further, by defining $\pi^*(y|x_g) = \frac{1}{Z(x_g)}\pi_{\text{ref}}(y|x_g)\exp(\frac{r(x_g,y)}{\beta})$, we can rewrite Equation (17) as:

$$\beta \min_{\pi\in\Pi} \max_{\alpha\in\Delta_{K-1}} \sum_{g=1}^{K} \alpha_g \Big( \mathbb{E}_{x_g\sim\mathcal{P}_{x_g}} \mathbb{E}_{y\sim\pi(\cdot|x_g)} \Big[ \log\big(\tfrac{\pi(y|x_g)}{\pi^*(y|x_g)}\big) - \log(Z(x_g))\Big]\Big) \tag{18}$$

$$=\beta \min_{\pi\in\Pi} \max_{\alpha\in\Delta_{K-1}} \sum_{g=1}^{K} \alpha_g \Big( \mathbb{E}_{x_g\sim\mathcal{P}_{x_g}} \Big[ D_{KL}\big(\pi(\cdot|x_g)||\pi^*(\cdot|x_g)\big) - \log(Z(x_g))\Big]\Big) \tag{19}$$

$$=\beta \min_{\pi\in\Pi} roL(\pi), \tag{20}$$

where we use

$$roL(\pi) := \max_{\alpha\in\Delta_{K-1}} \sum_{g=1}^{K} \alpha_g \Big( \mathbb{E}_{x_g\sim\mathcal{P}_{x_g}} \Big[ D_{KL}\big(\pi(\cdot|x_g)||\pi^*(\cdot|x_g)\big) - \log(Z(x_g))\Big]\Big).$$

It is not hard to show that $roL(\pi)$ is minimized by $\pi^*$. Indeed, let us consider any other policy $\pi \neq \pi^*$. Then, we have

$$roL(\pi) =\beta \max_{\alpha\in\Delta_{K-1}} \sum_{g=1}^{K} \alpha_g \Big( \mathbb{E}_{x_g\sim\mathcal{P}_{x_g}} \Big[ D_{KL}\big(\pi(\cdot|x_g)||\pi^*(\cdot|x_g)\big) - \log(Z(x_g))\Big]\Big) \tag{21}$$

$$\geq \beta \max_{\alpha\in\Delta_{K-1}} \sum_{g=1}^{K} \alpha_g \Big( \mathbb{E}_{x_g\sim\mathcal{P}_{x_g}} \Big[ -\log(Z(x_g))\Big]\Big) \tag{22}$$

$$= \beta \max_{\alpha\in\Delta_{K-1}} \sum_{g=1}^{K} \alpha_g \Big( \mathbb{E}_{x_g\sim\mathcal{P}_{x_g}} \Big[ D_{KL}\big(\pi^*(\cdot|x_g)||\pi^*(\cdot|x_g)\big) - \log(Z(x_g))\Big]\Big) \tag{23}$$

$$= roL(\pi^*), \tag{24}$$

where the inequality follows since $D_{KL}\big(\pi(\cdot|x)||\pi^*(\cdot|x)\big) \geq 0$. This implies $\pi^*$ minimises Equation (20). $\square$

Note that, we have proved that the optimal policy expression for Equation (12) aligns with the form of the optimal policy for standard (non-robust) KL-regularized reward maximization Equation (4).

**Proposition 3.2.** *Substituting the closed-form solution of the robust KL-regularized policy maximization problem (Equation* (10)*) into the robust reward maximization objective in Equation* (6) *leads to the same group robust DPO loss* $\mathcal{L}_{\text{GR−DPO}}$ *in Equation* (8) *.*

*Proof.* In [37], the DPO loss (see Equation (5)) is obtained by using the following reward loss:

$$\mathcal{L}_R(r;\mathcal{D}) = \mathcal{L}_R(r;\{\mathcal{D}_g\}_{g=1}^K) = -\sum_{g=1}^{K} \mathbb{E}_{(x_g,y_w,y_l)\sim\mathcal{D}_g} \big[\log\big(\sigma\big(r(x_g,y_w) - r(x_g,y_l)\big)\big)\big], \tag{25}$$

and replacing $r(\cdot,\cdot)$ with the relation

$$\pi^*(y|x) = \tfrac{1}{Z(x)}\pi_{\text{ref}}(y|x)\exp\big(\tfrac{r(x,y)}{\beta}\big), \tag{26}$$

which is the solution to

$$\pi^* = \max_{\pi\in\Pi} \sum_{g=1}^{K} \mathbb{E}_{(x_g\sim\mathcal{P}_{x_g},y\sim\pi(\cdot|x_g))} \Big[r_\phi(x_g,y)\Big] - \beta D_{KL}\Big[\pi(y|x_g)||\pi_{\text{ref}}(y|x_g)\Big]. \tag{27}$$

However, in our *group-robust* formulation, we replace Equation (25) with

$$\max_{g\in\mathcal{G}} \mathcal{L}_R(r;\{\mathcal{D}_g\}_{g=1}^K) = \max_{\alpha\in\Delta_{K-1}} \sum_{g=1}^{K} \alpha_g \Big( -\mathbb{E}_{(x_g,y_w,y_l)\sim\mathcal{D}_g} \Big[\log\big(\sigma\big(r(x_g,y_w) - r(x_g,y_l)\big)\big)\Big]\Big), \tag{28}$$

while Equation (27) remains the same. Alternatively, we can replace Equation (27) with its robust version, i.e.,

$$\pi^* = \max_{\pi \in \Pi} \min_{\alpha \in \Delta_{K-1}} \sum_{g=1}^{K} \alpha_g \left( \mathbb{E}_{(x_g \sim \mathcal{P}_{x_g}, y \sim \pi(\cdot|x_g))} \Big[ r_\phi(x_g, y) \Big] - \beta D_{KL} \Big[ \pi(y|x_g) || \pi_{\text{ref}}(y|x_g) \Big] \right),$$
(29)

however, we have already shown, in Lemma B.1, that this does not change the obtained policy-reward relation from Equation (26). Hence, this proves Proposition 3.2. $\qquad\square$

## B.3 Analysis of the Gradient

We elucidate the steps to obtain the expression for the loss gradient in Equation (11). We can simplify the gradient as follows:

$$\frac{\alpha_g^t \nabla_\theta l(\pi_{\theta^{t-1}}; (x_g, y_w, y_l))}{n_g}$$

$$= \frac{\alpha_g^t \nabla_\theta \log \Big( \sigma(\beta h_{\pi_{\theta^{t-1}}}(x_g, y_w, y_l)) \Big)}{n_g}$$

$$\overset{(i)}{=} \frac{\alpha_g^t}{n_g} \frac{\sigma'(\beta h_{\pi_{\theta^{t-1}}}(x_g, y_w, y_l))}{\sigma(\beta h_{\pi_{\theta^{t-1}}}(x_g, y_w, y_l))} \times [\beta h'_{\pi_{\theta^{t-1}}}(x_g, y_w, y_l)]$$

$$\overset{(ii)}{=} \frac{\beta \alpha_g^t}{n_g} \Big( 1 - \sigma(\beta h_{\pi_{\theta^{t-1}}}(x_g, y_w, y_l)) \Big) \times [h'_{\pi_{\theta^{t-1}}}(x_g, y_w, y_l)]$$

$$\overset{(iii)}{=} \frac{\beta \alpha_g^t}{n_g} \sigma\Big( -\beta h_{\pi_{\theta^{t-1}}}(x_g, y_w, y_l) \Big) \times [h'_{\pi_{\theta^{t-1}}}(x_g, y_w, y_l)]$$

$$\overset{(iv)}{=} \frac{\beta \alpha_g^t}{n_g} \sigma\Big( \beta \big( \log(\tfrac{\pi_{\theta^{t-1}}(y_l|x)}{\pi_{\text{ref}}(y_l|x)}) - \log(\tfrac{\pi_{\theta^{t-1}}(y_w|x)}{\pi_{\text{ref}}(y_w|x)}) \big) \Big) \times [h'_{\pi_{\theta^{t-1}}}(x_g, y_w, y_l)]$$

$$\overset{(v)}{=} \frac{\alpha_g^t}{n_g} \sigma\big( r_{\theta^{t-1}}(x_g, y_l) - r_{\theta^{t-1}}(x_g, y_w) \big) \times [\nabla_\theta \log \pi_{\theta^{t-1}}(y_w|x_g) - \nabla_\theta \log \pi_{\theta^{t-1}}(y_l|x_g)]$$

Here, (i) follows from the derivative of $\log(\cdot)$ function and denoting the derivative of sigmoid function $\sigma(\cdot)$ as $\sigma'(\cdot)$. (ii) and (iii) follow from the fact that for any $s \in \mathbb{R}$, $\sigma'(s) = 1 - \sigma(s) = \sigma(-s)$. Finally, (iv) and (v) follow using the definition of $h_\pi(x, y_w, y_l) = \log(\tfrac{\pi(y_w|x)}{\pi_{\text{ref}}(y_w|x)}) - \log(\tfrac{\pi(y_l|x)}{\pi_{\text{ref}}(y_l|x)})$, and substituting $r_\theta(x_g, y) = \beta \log(\tfrac{\pi_\theta(y|x_g)}{\pi_{\text{ref}}(y|x_g)}) - \beta \log Z(x_g)$ from Equation (4).

## B.4 Trading off worst-case vs. average performance

In the robust objective of Equation (8), we intend to minimize the worst-case loss. This often might adversely impact the average loss leading to bad average performance across the different groups. To mitigate this, we propose the following *robust trade-off* direct preference optimization objective for a specified policy $\pi$ and set of group weights $\mu_1, \ldots, \mu_K$:

$$\mathcal{L}_{\text{GR},\chi}(\pi) := (1 - \chi) \sum_{g=1}^{K} \mu_g \mathcal{L}_{\text{DPO}}(\pi, \mathcal{D}_g) + \chi \max_{g \in \mathcal{G}} \mathcal{L}_{\text{DPO}}(\pi, \mathcal{D}_g)$$

$$= (1 - \chi) \sum_{g=1}^{K} \mu_g \Big( -\mathbb{E}_{(x_g, y_w, y_l) \sim \mathcal{D}_g} \Big[ \log \Big( \sigma\big(\beta h_\pi(x_g, y_w, y_l)\big) \Big) \Big] \Big) +$$

$$\chi \max_{g \in \mathcal{G}} \Big( -\mathbb{E}_{(x_g, y_w, y_l) \sim \mathcal{D}_g} \Big[ \log \Big( \sigma\big(\beta h_\pi(x_g, y_w, y_l)\big) \Big) \Big] \Big).$$
(30)

Here, note that the input weights $\mu_g$ can be equal for all groups leading to a trade-off between the average and worst-case performance w.r.t. parameter $\chi$. Following Equation (8), this can be

equivalently recast into:

$$\mathcal{L}_{\text{GR},\chi}(\pi) = (1-\chi) \sum_{g=1}^{K} \mu_g \Big( - \mathbb{E}_{(x_g,y_w,y_l)\sim\mathcal{D}_g} \Big[ \log \Big( \sigma(\beta h_\pi(x_g, y_w, y_l)) \Big) \Big] \Big)$$

$$+ \chi \max_{\alpha \in \Delta_{K-1}} \sum_{g=1}^{K} \alpha_g \Big( - \mathbb{E}_{(x_g,y_w,y_l)\sim\mathcal{D}_g} \Big[ \log \Big( \sigma(\beta h_\pi(x_g, y_w, y_l)) \Big) \Big] \Big), \tag{31}$$

$$= \max_{\alpha \in \Delta_{K-1}} \sum_{g=1}^{K} ((1-\chi)\mu_g + \chi\alpha_g) \Big( - \mathbb{E}_{(x_g,y_w,y_l)\sim\mathcal{D}_g} \Big[ \log \Big( \sigma(\beta h_\pi(x_g, y_w, y_l)) \Big) \Big] \Big) \tag{32}$$

where $\Delta_{K-1}$ represents the $(K-1)$-dimensional simplex of probabilities. Hence, minimizing the loss in Equation (31) implies that one can implicitly find an optimal reward function using human group preference data that is robust without losing average performance, and obtain a policy that works effectively in terms of both average and worst-case performance:

$$\min_{\pi} \mathcal{L}_{\text{GR},\chi}(\pi). \tag{33}$$

As Equation (8) involves maximizing with respect to $\alpha$ and minimizing with respect to $\pi$, it forms a two-player game similar to the one considered in Section 3.1, where the policy $\pi$ and $\alpha$ act as opponents with inversely related payoffs. However, the contributions of the maximizing player ($\alpha$) on the final outcome are limited depending on parameters $\chi$ and $\mu$.

## C  IPO Simplification

Within the context of a finite data setting, it is common to record two responses $(y, y')$ for each prompt $x$. Based on these observations, there might be a tendency to inaccurately deduce that the preference distribution fulfills $p(y \succ y'|x) = 1$. Consequently, and assuming the Bradley-Terry model of preferences, the reward function $r$ has to satisfy $r(y, x) - r(y', x) \to \infty$. Further, given that DPO policy $\pi$ is directly dependent on the reward function (Equation (4)), it holds that $\frac{\pi(y'|x)}{\pi(y|x)} = 0$. In such a case, the policy overfits irrespective of the KL regularization factor. To circumvent this, [1] consider reward-free RLHF problem. Specifically, they propose preference optimization in the policy objective instead of reward optimization as follows:

$$\max_{\pi} \mathbb{E}_{x\sim\rho, y\sim\pi(\cdot|x), y'\sim\mu(\cdot|x)} \big[ \Psi(p(y \succ y')|x) \big] - \beta \text{KL}\big[ \pi(y|x) || \pi_{\text{ref}}(y|x) \big]. \tag{34}$$

Here, $\Psi : [0, 1] \to \mathbb{R}$ is a non-decreasing function, $\rho$ refers to the distribution of prompts and, $\mu$ refers to the competing behavior policy. Choosing $\Psi$ as the *identity* function, the optimal policy $\pi^*$ of Equation (34) has a similar expression to Equation (4) with the reward function $r(\cdot)$ substituted by the following preference function $p(\cdot)$,

$$p(y \succ \mu|x) = \beta \log \frac{\pi^*(y \mid x)}{\pi_{\text{ref}}(y \mid x)} + \beta \log Z(x). \tag{35}$$

Recalling the definition of $h_\pi(x, y, y') = \log(\frac{\pi(y|x)}{\pi_{\text{ref}}(y|x)}) - \log(\frac{\pi(y'|x)}{\pi_{\text{ref}}(y'|x)})$ from Section 2, it follows from Equation (35) that $\pi^*$ adheres to the following identity that equates the true preference with the policy's preference of $y$ over $y'$:

$$h_{\pi^*}(x, y, y') = \beta^{-1}(p(y \succ \mu|x) - p(y' \succ \mu|x)). \tag{36}$$

where $p(y \succ \mu|x) := \mathbb{E}_{\bar{y}\sim\mu(\cdot|x)}\big[ p(y \succ \bar{y}|x) \big]$. The equation motivates us to find a policy $\pi$ that minimizes the squared differences of two sides in Equation (36),

$$\mathcal{L}(\pi) = \mathbb{E}_{y,y'\sim\mu} \Big[ h_\pi(x, y, y') - \beta^{-1}(p(y \succ \mu|x) - p(y' \succ \mu|x)) \Big]^2. \tag{37}$$

For an empirical dataset where one observes if $y_w$ is more preferable to $y_l$, the policy minimization objective in Equation (37) becomes the IPO objective:

$$\mathcal{L}_{\text{IPO}}(\pi, \mathcal{D}) = \mathbb{E}_{(x,y_w,y_l)\sim\mathcal{D}} \Big[ h_\pi(x, y_w, y_l) - \frac{1}{2\beta} \Big]^2. \tag{38}$$

We can simplify Equation (38) for the IPO Loss using Log-linear policy class similar to Equation (9) as follows

$$L_{\text{IPO}}(\pi, \mathcal{D}) = \underset{(x, y_w, y_l) \sim \mathcal{D}}{\mathbb{E}} \left[ \left( \langle \phi(x, y_w) - \phi(x, y_l), \theta \rangle - \frac{1}{2\beta} \right)^2 \right]. \tag{39}$$

Contrary to DPO, the IPO Loss results in a problem formulation equivalent to linear regression for the linear bandit setting. This leads us to obtain a closed form analytical solution for $\theta$

$$\hat{\theta}_{IPO} = \frac{1}{2\beta} (X^T X)^{-1} X^T \mathbf{1}, \tag{40}$$

where $\mathbf{1} = \{1\}^N$. Here, $X \in \mathbb{R}^{d \times N}$. In particular, each row of $X$ is the difference of the preferred and least preferred feature vectors $\phi(x, y_w) - \phi(x, y_l)$ for $(x, y_w, y_l) \in \mathcal{X} \times \mathcal{Y} \times \mathcal{Y}$

$$X = \begin{bmatrix} \phi(x_1, y_1) - \phi(s_1, y_1') \\ \phi(x_2, y_2) - \phi(x_2, y_2') \\ \vdots \\ \phi(x_n, y_n) - \phi(x_n, y_n') \end{bmatrix}.$$

When re-written in this form it is trivial to see that the stability of the IPO Loss depends upon the rank of the matrix $X$.

**Group IPO:** We begin by conducting experiments with IPO considering the existence of closed form solution for this particular setting. In particular, given a set of preference data $\mathcal{D} = \{\mathcal{D}_1, \mathcal{D}_2\}$ from either groups with varying ratio, one aims to find an optimal $\theta$ that minimizes the group IPO loss,

$$L_{\text{IPO}}(\pi, \mathcal{D}) = \underset{(x, y_w, y_l) \sim \mathcal{D}_1}{\mathbb{E}} \left[ \left( \langle \phi(x, y_w, 0) - \phi(x, y_l, 0), \theta \rangle - \frac{1}{2\beta} \right)^2 \right] \tag{41}$$

$$+ \underset{(x, y_w, y_l) \sim \mathcal{D}_2}{\mathbb{E}} \left[ \left( \langle \phi(x, y_w, 1) - \phi(x, y_l, 1), \theta \rangle - \frac{1}{2\beta} \right)^2 \right]. \tag{42}$$

Concisely, one can also write this as,

$$L_{\text{IPO}}(\pi, \mathcal{D}) = \underset{(x, y_w, y_l, g) \sim \mathcal{D}}{\mathbb{E}} \left[ \left( \langle \phi(x, y_w, g) - \phi(x, y_l, g), \theta \rangle - \frac{1}{2\beta} \right)^2 \right]. \tag{43}$$

We assume that the feature vectors for each group is known, but the reward parameter $\theta$ is unknown. So, given the preference data with group information and the group dependent feature vectors, one aims to learn an optimal $\theta$ that balances both groups by minimizing the above loss. The solution to this will still be the closed form solution expressed in Equation (40) with feature matrix $X$ consisting group dependent features.

**Group Robust IPO:** It is straightforward to see that in such a setting, the group with higher number of preference data will have an unfair advantage as they contribute more to the loss. Hence, a robust approach needs to be considered as follows:

$$roL_{\text{IPO}}(\pi, \mathcal{D}) = \max_{\sum_g \alpha_g = 1} \underset{(x, y_w, y_l, g) \sim \mathcal{D}}{\mathbb{E}} \left[ \alpha_g \left( \langle \phi(x, y_w, g) - \phi(x, y_l, g), \theta \rangle - \frac{1}{2\beta} \right)^2 \right]. \tag{44}$$

One can use, Algorithm 1 to find the optimal $\theta$ for Equation (44). But considering the weighted regression form of the loss, one can use the following simplified algorithm Algorithm 2. Note that the last step, inadvertently leads to a weighted regression whose solution can be obtained in closed form.

# D Additional Experiments and Details

In this section, we discuss additional experimental and training details. We use the following hyperparameters for the synthetic experiments. The importance sampling methods use the same hyperparameters as the corresponding vanilla ones. Further, we note that there is no learning rates for IPO and GR-IPO as we use the closed-form solution detailed in Section 4.1 for updates.

---

**Algorithm 2** Policy Optimization for Robust IPO in Linear Bandits

---

1: **Initialize:** Step size $\eta_\alpha$ for group weights $\alpha$, initial weights $\theta^{(0)}$ of the policy and weights over each group $\alpha^{(0)}$
2: **Input:** Dataset $\mathcal{D}$ with size $N = |\mathcal{D}|$, group size $N_g$ for $g = \{1, 2, \cdots, K\}$
3: **for** $t = 1, \ldots, T$ **do**
4:     Calculate group loss $l_g$ for each group $g$ on $\mathcal{D}_g$
5:     $\alpha' \leftarrow \alpha^{t-1}; \alpha'_g \leftarrow \alpha'_g \exp(\eta_G l_g)$      // Update weights for group g
6:     $\alpha^{(t)} \leftarrow \alpha' / \sum_{g'} \alpha'_{g'}$      // Renormalize $\alpha$
7:     $\theta^t \leftarrow \arg\min_\theta \mathbb{E}_{(x,y_w,y_l,g)\sim\mathcal{D}} \left[ \alpha_g^{(t)} \big( \langle \phi(x,y_w,g) - \phi(x,y_l,g), \theta \rangle - \frac{1}{2\beta} \big)^2 \right]$
8: **end for**
9: **Return:** Output the robust policy $\pi(\theta^t)$

---

| Training Type | Learning Rate | $\beta$ | Step Size |
|---|---|---|---|
| DPO | 0.9 | 1 | - |
| IPO | - | 0.1 | - |
| GR-DPO | 0.9 | 1 | 0.5 |
| GR-IPO | - | 0.1 | 0.01 |

Table 1: Hyperparameters for synthetic experiments.

## D.1 Synthetic Experiments - Experiments setup

Here we provide a full definition of our synthetic experimental setup discussed in Section 5.

We adapt the standard (non-group based) experimental setup proposed by [26] for the group preferences setting by incorporating group information into the reward function $r : \mathcal{X} \times \mathcal{Y} \times \mathcal{G} \to \mathbb{R}$. Here, $\mathcal{X}$ represents a two-dimensional state space $[0, 1] \times [0, 1]$, $\mathcal{Y}$ denotes a discrete action space $\{0, 1, 2, 3, \ldots, n\}$, and $\mathcal{G}$ signifies a discrete group space $\{0, 1, 2, \ldots, K\}$. The reward function, defined by the group-dependent feature vector $\phi(x, y, g)$ and parameter vector $\theta_g$, is given as:

$$r(x, y, g) := \langle \phi(x, y, g), \theta_g \rangle. \tag{45}$$

Further, we consider $n = 7$, $K = 2$ and denote $x := (x_0, x_1) \in \mathcal{X}$. The feature vector $\phi(x, y, g) := (\phi_0(x, y, g), \phi_1(x, y, g), \phi_2(x, y, g), \phi_3(x, y, g))$ and parameters $\theta_g \ \forall g \in \mathcal{G}$ are defined as follows:

$$\phi_i(x, y, g) := \begin{cases} \left(\frac{y}{n+1} + 1\right) \cdot \cos(x_{\lfloor i/2 \rfloor} \cdot \pi) & \text{if } i\%2 = g \\ \left(\frac{1}{\frac{y}{n+1}+1}\right) \cdot \sin(x_{\lfloor i/2 \rfloor} \cdot \pi) & \text{otherwise} \end{cases}, \quad \theta_0 := (1, 3, 1, 3), \theta_1 := (3, 1, 3, 1).$$

By design, this parameterization ensures a coordinate-flipped relationship between groups, effectively mirroring one group to the other. Also, we study two other feature parameterizations and include their experimental results in Appendix D.3.

## D.2 Synthetic Experiments - Preferences data generation

In our synthetic experiments, we considered three scenarios: **(i)** Groups are imbalanced in size, **(ii)** preferences distribution, and **(iii)** both of the above. The size imbalance is generated by using ratios $0.2 : 0.8$, respectively. Instead, to generate imbalance in terms of preferences we proceed as follows. In scenario **(i)**, after randomly selecting a state $x$ within group $g$ and an action $y_1$, a second action $y_2$ is randomly chosen (groups have the same distribution over responses). Instead, in scenarios **(ii)** and **(iii)**, for one group $y_2$ is the action most distant from $y_1$, while for the other, it is the closest (groups have different distributions over responses). In both methods, we calculate the rewards $r(x, y_1, g)$ and $r(x, y_2, g)$, designating the action with the higher reward as preferred.

Here, we provide the resulting plots for scenarios **(i)** and **(ii)**.

### Trading off worst-case vs. Average performance

We perform ablation studies for various values of $\chi$ trading-off between the average and worst-case performance. We observe in Figure 6, that the max validation loss decreases while moving from

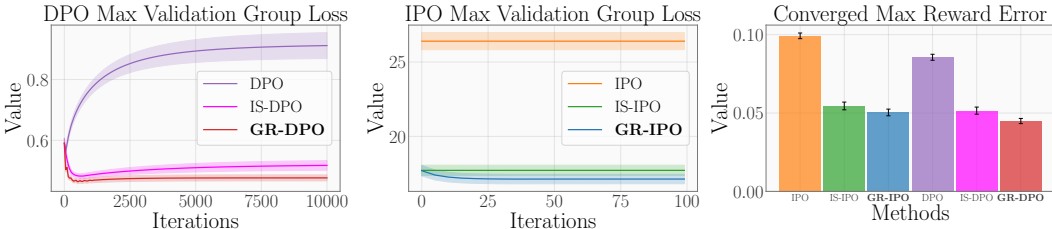

Figure 4: Algorithm 1 (GR-DPO and GR-IPO) leads to a lower worst-case validation loss and reward error compared to importance sampling and vanilla methods. Results refer to the scenario in which groups have different sizes but same responses' distribution. Note that the gap between Algorithm 1 and importance sampling is smaller than in Figure 4. This is expected considering that the primary difference between groups arises from data imbalance, which is handled by importance sampling.

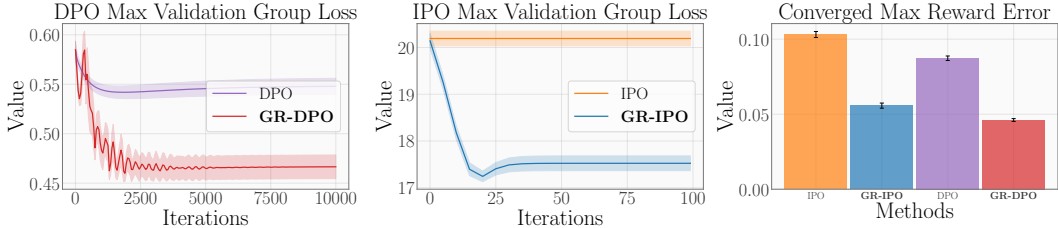

Figure 5: Algorithm 1 (GR-DPO and GR-IPO) leads to a lower worst-case validation loss and reward error compared to the non-robust vanilla methods. Results refer to the scenario in which groups have same sizes but different responses' distribution. Unlike the setups of Figure 4 and Figure 2 importance sampling has no effect here (it coincides with vanilla DPO/IPO since groups have the same sizes).

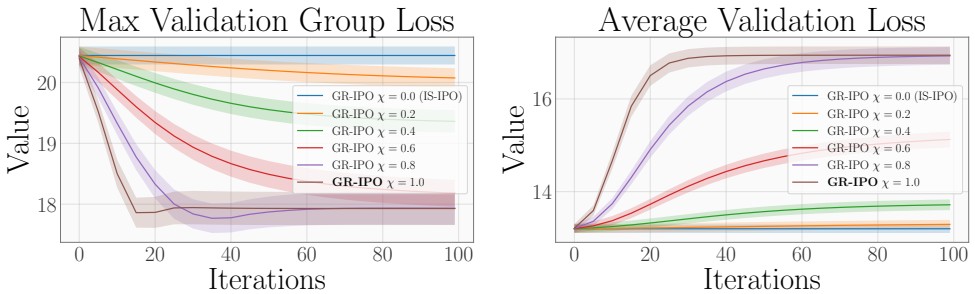

Figure 6: Ablation study for the trade-off parameter $\chi$ in the synthetic experimental setup. Results refer to the scenario in which groups have different sizes and different responses' distribution. Note that increasing $\chi$ improves worst group performance at the expense of average performance.

$\chi = 0$ to $\chi = 1$, where $\chi = 0$ corresponds to importance sampling with group weights $\mu_1, \cdots, \mu_g$ mapping to importance sampling weights (see Equation (30)) and $\chi = 1$ corresponds to GR-IPO. Further, we plot the average validation loss which increases while moving from $\chi = 0$ to $\chi = 1$, demonstrating the trade-off between average and worst-case performance. Note that, GR-IPO aptly increases the average loss (as expected) in order to reduce the worst group loss.

### D.3 Synthetic Experiments - Additional feature parametrizations

We present further experiments on synthetic preference data, using different configurations of the $\phi(x, y, g)$ and $\theta_g$ vectors characterising the group-specific reward distributions $r(x, y, g) = \langle \phi(x, y, g), \theta_g \rangle$.

With the same action space $n = 7$ and group space $K = 2$, we consider *same* and *flipped* configurations of $\phi(x, y, g) = (\phi_0, \phi_1, \phi_2, \phi_3)$, as follows:

**Same:** Here, the feature vectors are the same irrespective of group $g$ and reward vectors are coordinate-flipped.

$$\phi_i(x,y,g) := \begin{cases} \left(\frac{y}{n+1} + 1\right) \cdot \cos(x_{\lfloor i/2 \rfloor} \cdot \pi) & \text{if } i\%2 = 0 \\ \left(\frac{1}{\frac{y}{n+1}+1}\right) \cdot \sin(x_{\lfloor i/2 \rfloor} \cdot \pi) & \text{otherwise} \end{cases}, \quad \theta_0 := (1,3,1,3), \theta_1 := (3,1,3,1).$$

**Flipped:** Here, the feature vectors include alternating $\sin, \cos$ terms with swapped order for $g = \{0, 1\}$, with alternating coefficients. And the reward vectors are coordinate-flipped as before.

$$\phi_i(x,y,g) := \begin{cases} \left(\frac{y}{n+1} + 1\right)^{-1 \cdot \mathbb{1}\left[i\%2=1\right]} \cdot \cos(x_{\lfloor i/2 \rfloor} \cdot \pi) & \text{if } i\%2 = g \\ \left(\frac{y}{n+1} + 1\right)^{-1 \cdot \mathbb{1}\left[i\%2=1\right]} \cdot \sin(x_{\lfloor i/2 \rfloor} \cdot \pi) & \text{otherwise} \end{cases}$$

The experimental results are shown in Figures 7, 8 and 9 for the *same* experiment, and Figures 10, 11 and 12 for the *flipped* experiment.

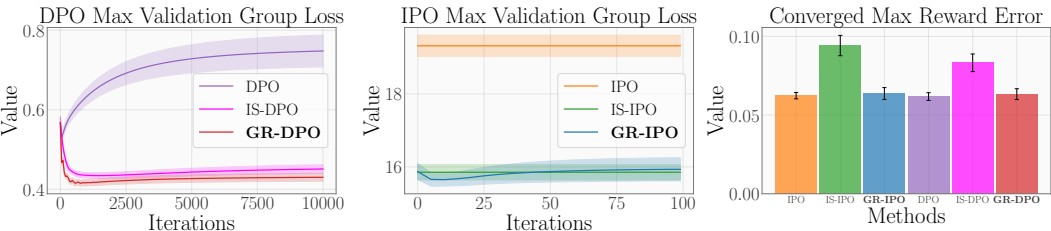

Figure 7: In this experiment, we consider *same* feature vectors on *even* groups imbalanced in size. We observe similar performance as in Figure 4 where we consider the same setting with swapped feature vectors for groups. GR-DPO slightly improves over importance sampling DPO, however, GR-IPO exactly matches the performance of importance sampling. And this is expected considering that the groups have the same level of difficulty and differ only in terms of data size.

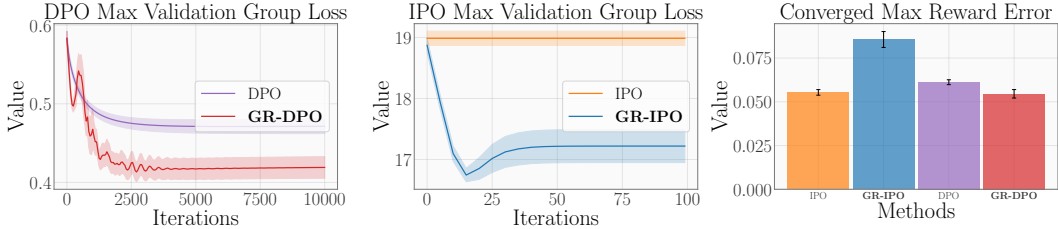

Figure 8: In this experiment, we consider *same* feature vectors on *uneven* groups balanced in size. Here, GR-DPO/GR-IPO outperforms corresponding importance sampling methods in terms of worst-case validation loss, but, GR-IPO tends to overfit. This is reflected in the reward errors, where GR-IPO performs worse than IPO. However, overall, GR-DPO outperforms all other methods in terms of worst-case reward errors.

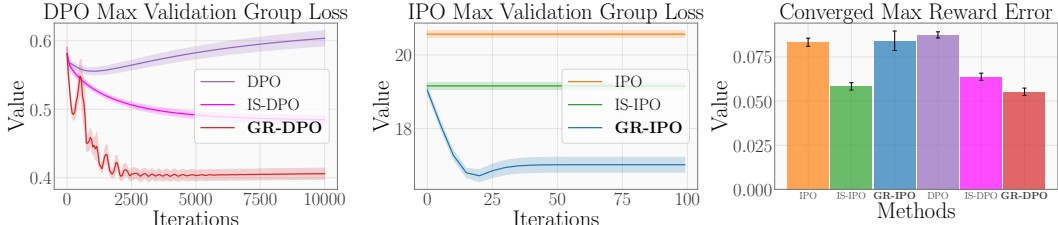

Figure 9: In this experiment, we consider *same* feature vectors on *uneven* groups imbalanced in size. We observe similar performance as in Figure 8 where we consider the same features for groups but with balanced data. Here, DPO is overfitting and GR-DPO outperforms both DPO and importance sampling DPO. IPO is stable, considering it has a closed form solution. And in terms of worst-case validation loss, GR-IPO tends to overfit, even though it outperforms IPO and importance sampling IPO. The overfitting is more evident in reward errors, where GR-IPO performs worse than IPO and importance sampling IPO. However, overall, GR-DPO outperforms all other methods in terms of worst-case reward errors.

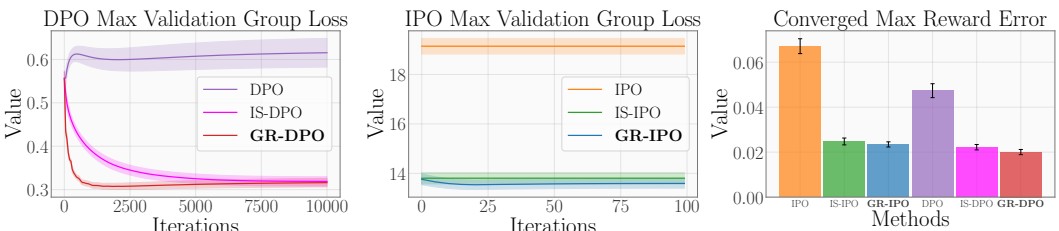

Figure 10: In this experiment, we consider *flipped* feature vectors on *even* groups imbalanced in size. We observe very similar performance as in Figure 4 where GR-DPO/GR-IPO slightly improves over importance sampling DPO/IPO.

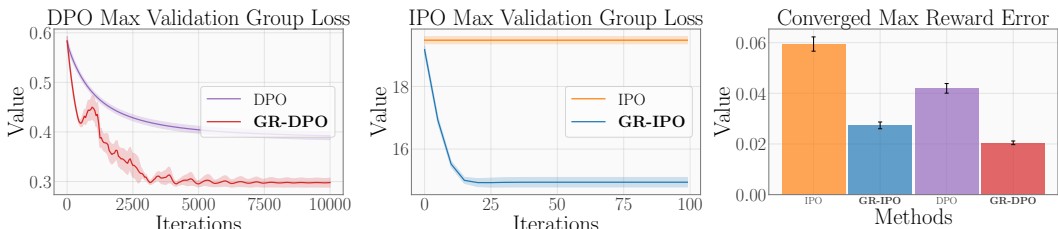

Figure 11: In this experiment, we consider *flipped* feature vectors on *uneven* groups balanced in size. Here, GR-DPO/GR-IPO outperforms corresponding vanilla methods in terms of worst-case validation loss and GR-DPO outperforms all other methods in terms of worst-case reward errors.

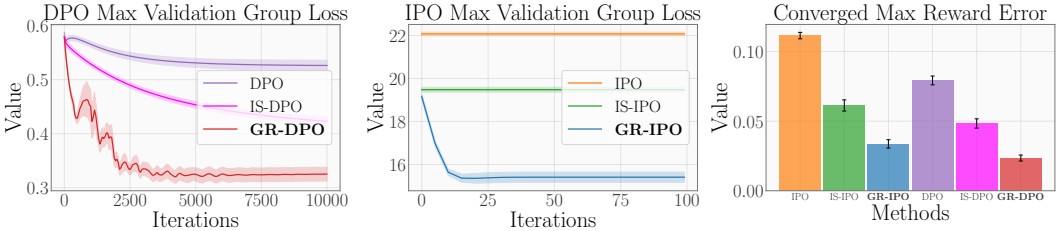

Figure 12: In this experiment, we consider *flipped* feature vectors on *uneven* groups imbalanced in size. Here, we observe a similar performance to that of Figure 2 where GR-DPO/GR-IPO outperforms corresponding vanilla and importance sampling methods in terms of worst-case validation loss. And, GR-DPO outperforms all other methods in terms of worst-case reward errors.

### D.4 Global Opinion Data Experiments - Setup and Additional Plots

We use the following prompt type for both SFT and DPO training: "Opinion of people in (country-name) on: (question). Please select the best response: A) Choice-1, B) Choice-2, ...". The target response is just the letter 'A', 'B', etc. The training data size from each country are as follows: Nigeria-572, Egypt- 570, India-376, China-309, Japan-712. Further, the data is split as $80\%$ for training, $10\%$ for validation and $10\%$ for testing. We run the SFT training with learning rate $10^{-4}$ for one epoch over the training data on pre-trained Gemma-2B model. We use this SFT trained model as the reference model for training IPO and GR-IPO. For the IPO training, the optimal hyperparameters were learning rate $3 * 10^{-5}$, and $\beta = 0.01$. For the GR-IPO training, we use the same $\beta$ but the optimal learning rate and the exponential step size were $6 * 10^{-5}$ and $5 * 10^{-7}$. For both IPO/GR-IPO training we use AdamW [27] optimizer with adaptive learning rates decreasing by a factor of 10 if there is no improvement in terms of loss (average group loss for IPO/ worst group loss for GR-IPO) for $4k$ iterations on a validation set. Further, for GR-IPO, the exponential step size is decreased by a factor of 2 whenever learning rate is reduced. This adaptive learning rates and exponential step sizes ensures stability in training and leads to convergence. We then evaluate both the methods based on the worst group loss and accuracy. Here, the accuracy refer to the percentage of prompt, winning response and losing response correctly ordered by the learned preference function (Equation (35)).

All experiments were run on a single node of A100 SXM4 machine with 40GB GPU memory, 30 CPU cores, 200GB RAM, and 525GB SSD memory. Further, for each seed, the execution time of the experiments is approximately 3-5 hours until convergence. We also provide the exact hyperparameters used in the table below.

| Training Type | Learning Rate | $\beta$ | Step Size | Optimizer |
|---|---|---|---|---|
| SFT | $10^{-4}$ | - | | RmsProp |
| IPO | $3 \times 10^{-5}$ | 0.01 | - | AdamW |
| GR-IPO | $6 \times 10^{-5}$ | 0.01 | $5 \times 10^{-7}$ | AdamW |

Table 2: Hyperparameters for SFT, IPO, and GR-IPO training.

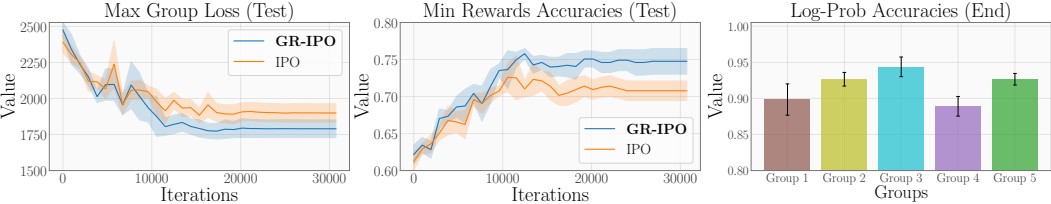

Figure 13: Global opinion data: Evolution of worst-case loss, reward accuracy during IPO and GR-IPO training. Notably, both IPO and GR-IPO improve their accuracy. However, GR-IPO achieves better worst-case alignment performance. In the right, we plot the end of training log-prob. accuracies for GR-IPO.

# E   Convergence Proofs for Different Sampling Strategies

In this section, we study the convergence rates for different sampling strategies including the proposed Algorithm 1.

## E.1   Convergence Proof for Algorithm 1

In this section, we prove the convergence of the proposed Algorithm 1 in terms of the error of the average iterate $\pi_{\bar{\theta}(1:T)}$,

$$\epsilon_T = \mathcal{L}_{\mathrm{GR}}(\pi_{\bar{\theta}(1:T)}) - \min_{\theta \in \Theta} \mathcal{L}_{\mathrm{GR}}(\pi_\theta), \tag{46}$$

as stated in Proposition 4.1.

**Proposition 4.1.** *Suppose that the loss $l(\cdot; (x_g, y, y'))$ is non-negative, convex, $B_\nabla$−Lipschitz continuous, and bounded by $B_l$ for all $(x_g, y, y') \in \mathcal{X} \oplus \mathcal{G} \times \mathcal{Y} \times \mathcal{Y}$ and $\|\theta\|_2 \leq B_\Theta$ for all $\theta \in \Theta$ with convex $\Theta \subset \mathbb{R}^d$. The error of the average iterate of Algorithm 1, i.e., $\pi_{\bar{\theta}(1:T)} = \frac{1}{T} \sum_{t=1}^{T} \theta^t$, satisfies*

$$\mathbb{E}[\mathcal{L}_{\mathrm{GR}}(\pi_{\bar{\theta}(1:T)})] - \min_{\theta \in \Theta} \mathcal{L}_{\mathrm{GR}}(\pi_\theta) = \mathcal{O}(T^{-1/2}).$$

*Proof.* We build our proof based upon the online mirror descent algorithm's regret bound in [32, Eq. 3.1]. [32]'s theorem considers the following saddle-point stochastic optimization problem,

$$\min_{\theta \in \Theta} \max_{\alpha \in \Delta_{K-1}} \left\{ \phi(\theta, \alpha) = \sum_{g=1}^{K} \alpha_g \mathbb{E}[F_g(\theta, \xi)] \right\}. \tag{47}$$

Denote $f_g(\theta) = \mathbb{E}[F_g(\theta, \xi)]$. In our problem, we consider,

$$\xi := (x_g, y, y') \sim \mathcal{D}, \tag{48}$$

which is equivalent to

$$\xi := (x_g, y, y') \sim \sum_{g=1}^{K} \frac{N_g}{N} \mathcal{D}_g, \tag{49}$$

and we define $F_{g'}(\theta, (x_g, y, y')) := \frac{N}{N_g} \mathbf{1}_{[g=g']} l(\theta; (x_g, y, y'))$. Then, the expectation

$$
\begin{aligned}
\mathbb{E}_\xi[F_g(\theta, \xi)] &= \sum_{g'=1}^{K} \frac{N_{g'}}{N} \mathbb{E}_{\mathcal{D}_{g'}}[F_g(\theta, \xi)|g = g'] \\
&= \frac{N_g}{N} \mathbb{E}_{\mathcal{D}_g}[F_g(\theta, \xi)|g = g'] \\
&= \frac{N_g}{N} \mathbb{E}_{\mathcal{D}_g}\left[\frac{N}{N_g} l(\theta; (x_g, y, y'))|g = g'\right] \\
&= \mathbb{E}_{\mathcal{D}_g}[l(\theta; (x_g, y, y'))|g = g'] \\
&= f_g(\theta)
\end{aligned}
$$

**Mirror Descent mapping.** We begin by mapping Algorithm 1 to the general mirror descent framework of [32].

Further, we denote the gradients of the objective $\phi(\theta, \alpha)$ w.r.t. $\theta$ and $\alpha$ as

$$\partial\phi(\theta, \alpha) = (\partial_\theta \phi(\theta, \alpha), -\partial_\alpha \phi(\theta, \alpha)).$$

Here, the negative sign for the gradient w.r.t. $\alpha$ indicates that we perform maximization w.r.t. $\alpha$. Let $\Phi(\theta, \alpha, \xi) := \sum_{g=1}^{K} \alpha_g[F_g(\theta, \xi)]$. Then the stochastic subgradients for $\phi(\theta, \alpha)$ for a given $\xi$ is as follows:

$$\partial\Phi(\theta, \alpha, \xi) = (\partial_\theta \Phi(\theta, \alpha, \xi), -\partial_\alpha \Phi(\theta, \alpha, \xi)).$$

Simplifying the gradients we obtain the following for $\xi = (x_g, y, y')$:

$$\partial \Phi(\theta, \alpha, \xi) = \begin{bmatrix} \partial_\theta \sum_{g'=1}^K \alpha_{g'}[F_{g'}(\theta, \xi)] \\ -\partial_\alpha \sum_{g'=1}^K \alpha_{g'}[F_{g'}(\theta, \xi)] \end{bmatrix}$$

$$= \begin{bmatrix} \partial_\theta \sum_{g'=1}^K \alpha_{g'}[\frac{N}{N_{g'}} \mathbf{1}_{[g=g']} l(\theta; (x_g, y, y'))] \\ -\partial_\alpha \sum_{g'=1}^K \alpha_{g'}[\frac{N}{N_{g'}} \mathbf{1}_{[g=g']} l(\theta; (x_g, y, y'))] \end{bmatrix}$$

$$= \begin{bmatrix} \frac{N\alpha_g}{N_g} \nabla_\theta l(\theta; (x_g, y, y')) \\ -\left(0, \cdots, \frac{N}{N_g} l(\theta; (x_g, y, y')), \cdots, 0\right) \end{bmatrix}.$$

Based on the above gradient expressions, we can map the updates of $\theta^t$ and $\alpha^t$ in Algorithm 1, to the general mirror descent update rule:

$$\zeta^{t+1} = (\theta^{t+1}, \alpha^{t+1}) = P_{(\zeta^t)}(\gamma^t \partial \Phi(\theta^t, \alpha^t, \xi)). \tag{50}$$

Here, we use the compact notation $\zeta = (\theta, \alpha)$. Further, $\gamma^t$ denotes the step-size at time $t$ of the algorithm and $P_\zeta(\cdot)$ is the prox-mapping corresponding to the mirror descent algorithm defined as follows:

$$P_\zeta(\nu) = \underset{\zeta' \in (\Theta \times \Delta_{K-1})}{\arg\min} \left\{ \nu^\top (\zeta' - \zeta) + V(\zeta, \zeta') \right\}. \tag{51}$$

Here, $V(\cdot, \cdot)$ is the prox-function associated with the distance-generating function (Bregmann Divergence) $\omega(\cdot)$. For our setup in Algorithm 1, we define the following combined distance-generating function over $\zeta \in (\Theta \times \Delta_{K-1})$:

$$\omega(\zeta) = \frac{\omega_\theta(\theta)}{2D_{\omega_\theta,\Theta}^2} + \frac{\omega_\alpha(\alpha)}{2D_{\omega_\alpha,\Delta_{K-1}}^2}, \tag{52}$$

where $\omega_\theta(\theta) = \frac{1}{2}\|\theta\|^2$, $\omega_\alpha(\alpha) = \sum_{g=1}^K \alpha_i \ln \alpha_i$, $D_{\omega_\theta,\Theta} := (\max_{\theta \in \Theta} \omega_\theta(\theta) - \min_{\theta \in \Theta} \omega_\theta(\theta))^{1/2}$, and $D_{\omega_\alpha,\Delta_{K-1}} := (\max_{\alpha \in \Delta_{K-1}} \omega_\alpha(\alpha) - \min_{\alpha \in \Delta_{K-1}} \omega_\alpha(\alpha))^{1/2}$. Also, the corresponding prox-functions individually for $\theta$ and $\alpha$ would be $V_\theta(\theta, \theta') = \frac{1}{2}\|\theta - \theta'\|_2^2$ and $V_\alpha(\alpha, \alpha') = \sum_{g=1}^K \alpha_i' \ln \frac{\alpha_i'}{\alpha_i}$. It can be shown that the above prox-functions $V_\theta$ and $V_\alpha$ satisfy the following inequalities ([32]):

$$\max_{\theta \in \Theta} V_\theta(\theta', \theta) \le D_{\omega_\theta,\Theta}^2, \quad \max_{\alpha \in \Delta_{K-1}} V_\alpha(\alpha', \alpha) \le D_{\omega_{\alpha'},\Delta_{K-1}}^2. \tag{53}$$

Due to the above definitions, the corresponding prox-mapping $P(\cdot)$ corresponds to gradient descent w.r.t. $\theta$ and exponentiated gradient ascent w.r.t. $\alpha$, as defined in Algorithm 1. Moreover, due to the definition of $\omega$ in Equation (52), the combined prox function $V(\cdot, \cdot)$ over $\zeta$ would satisfy

$$\max_{\zeta \in \Theta \times \Delta_{K-1}} V(\zeta', \zeta) \le 1. \tag{54}$$

For further details regarding prox-function, kindly refer to its usage in [32].

**Bounding the error.** According to the introduced notation, the approximation error $\epsilon_T$ can be defined and bounded as:

$$\epsilon_T = \max_{\alpha \in \Delta_{K-1}} \phi(\tilde{\theta}^{1:T}, \alpha) - \min_{\theta \in \Theta} \max_{\alpha \in \Delta_{K-1}} \phi(\theta, \alpha)$$

$$\le \max_{\alpha \in \Delta_{K-1}} \phi(\tilde{\theta}^{1:T}, \alpha) - \min_{\theta \in \Theta} \phi(\theta, \tilde{\alpha}^{1:T})$$

$$= \epsilon_\phi(\tilde{\zeta}^{1:T})$$

To bound $\epsilon_\phi(\tilde{\zeta}^{1:T})$, we invoke the result from [32, Equation 3.23] for mirror-descent convergence for max-min problems.

$$\epsilon_\phi(\tilde{\zeta}^{1:T}) \le 2\sqrt{10 \frac{R_\theta^2 M_\theta^2 + \ln(K) M_\alpha^2}{T}}. \tag{55}$$

Further, $M_\theta$ and $M_\alpha$ correspond to the following upper bounds to the maximum of the expected norms of $F_g(\theta, \xi)$ and $\nabla_\theta F_g(\theta, \xi)$,

$$
\begin{aligned}
\max_{1 \leq g \leq K} \mathbb{E}\|\nabla F_g(\theta, \xi)\|^2 = \max_{1 \leq g \leq K} \mathbb{E}\left\|\frac{N}{N_g}\mathbf{1}_{[g=g']}\nabla l(\theta; (x_g, y, y'))\right\|^2 &= \max_{1 \leq g \leq K} \frac{N_g}{N}\frac{N^2}{N_g^2}\|\nabla l(\theta; (x_g, y, y'))\|^2 \\
&= \max_{1 \leq g \leq K} \frac{N}{N_g}\|\nabla l(\theta; (x_g, y, y'))\|^2 \\
&\leq B_\nabla^2 \max_{g \in \mathcal{G}} \frac{N}{N_g} \\
&= B_\nabla^2 \frac{N}{\min_{g \in \mathcal{G}} N_g} \\
&= M_\theta^2
\end{aligned}
$$

Similarly,

$$
\begin{aligned}
\mathbb{E}\max_{1 \leq g \leq K}\|F_g(\theta, \xi)\|^2 = \mathbb{E}\max_{1 \leq g \leq K}\left\|\frac{N}{N_g}\mathbf{1}_{[g=g']}l(\theta; (x_g, y, y'))\right\|^2 &\leq \sum_{g=1}^{K} \frac{N_g}{N}\frac{N^2}{N_g^2}\|l(\theta; (x_g, y, y'))\|^2 \\
&= \sum_{g=1}^{K} \frac{N}{N_g}\|l(\theta; (x_g, y, y'))\|^2 \\
&\leq KB_l^2 \max_{g \in \mathcal{G}} \frac{N}{N_g} \\
&= KB_l^2 \frac{N}{\min_{g \in \mathcal{G}} N_g} \\
&= M_\alpha^2
\end{aligned}
$$

Here, we have used $\|\nabla_\theta l(\theta; (x_g, y))\| \leq B_\nabla$ and $\|l(\theta; (x_g, y))\| \leq B_l$. Here, $\lambda_\theta$, $\lambda_\alpha$ correspond to the strong convexity parameters of the distance generating functions $\omega_\theta(\theta)$ and $\omega_\alpha(\alpha)$ respectively. For our given functions, $\omega_\theta(\theta) = \frac{1}{2}\|\theta\|^2$ and $\omega_\alpha(\alpha) = \sum_{g=1}^{K} \alpha_i \ln \alpha_i$, both $\lambda_\theta = \lambda_\alpha = 1$ ([32]). Let, $R_\theta^2 = \frac{D_{\omega_\theta}^2}{\lambda_\theta} = B_\Theta^2 = (\max_\theta \|\theta\|_\theta - \min_\theta \|\theta\|_\theta^2)$, obtaining the overall error bound.

$$
\epsilon_\Phi(\tilde{\zeta}^{1:T}) \leq 2\sqrt{10\left(\frac{N}{\min_{g \in \mathcal{G}} N_g}\right)\frac{B_\Theta^2 B_\nabla^2 + KB_l^2 \ln K}{T}}. \tag{56}
$$

$\square$

**Lemma E.1.** *For the log-linear policy class parameterized with respect to $\theta$, the DPO loss function $l(\pi_\theta; \cdot) = \log\left(\sigma(\beta h_{\pi_\theta}(\cdot))\right)$ (see Equation (5)) is convex and Lipschitz continuous in $\theta$. Consequently, Proposition 4.1 applies to this case.*

*Proof.* We want to show, $l(\theta; (x, y_w, y_l))$ is convex in $\theta$ and $B_\phi$-Lipschitz in $\theta$ for the log-linear policy class defined as follows:

$$
\pi_\theta(y \mid x) = \frac{\exp \theta^T \phi(x, y)}{\sum_{y \in \mathcal{Y}} \exp \theta^T \phi(x, y)}. \tag{57}
$$

Here $\theta$ belongs to a convex set $\Theta$ satisfying $\|\theta\| \leq B_\Theta$ and $\phi(x, y)$ denotes the feature vector such that $\|\phi(x_g, y_w) - \phi(x_g, y_l)\| \leq B_\phi$. In conjunction with a uniform reference policy $\pi_{\text{ref}}$, the loss function $l(\theta; (x_g, y_w, y_l))$ for the log-linear policy class is as follows:

$$
l(\theta; (x_g, y_w, y_l)) = -\log\left(\sigma\left(\beta\langle\phi(x_g, y_w) - \phi(x_g, y_l), \theta\rangle\right)\right). \tag{58}
$$

**Algorithm 3** Mirror Descent for Group Robust Preference Optimization (GRPO)

---

1: **Initialize:** Step size $\eta_\alpha$ for group weights $\alpha$, step size $\eta_\theta$ for policy $\pi$ with weights $\theta$, initial weights $\theta^{(0)}$ of the policy and weights over each group $\alpha^{(0)}$, Projection operator $P_\Theta$
2: **Input:** Dataset $\mathcal{D}$ with size $N = |\mathcal{D}|$, group size $N_g$ for $g = \{1, 2, \cdots, K\}$, loss $l(\pi_\theta; \cdot)$
3: **for** $t = 1, \ldots, T$ **do**
4:     $\alpha' \leftarrow \alpha^{(t-1)}$
5:     $g \sim \text{Uniform}(1, \cdots, K)$
6:     $(x_g, y_w, y_l) \sim \mathcal{D}_g$
7:     $\alpha'_g \leftarrow \alpha'_g \exp \eta_\alpha \big( l(\pi_{\theta^{(t-1)}}; (x_g, y_w, y_l)) \big)$    // Update weights for group g
8:     $\alpha^{(t)} \leftarrow \alpha' / \sum_{g'} \alpha'_{g'}$     // Renormalize $\alpha$
9:     $\theta^{(t)} \leftarrow P_\Theta \Big( \theta^{(t-1)} - \eta_\theta \big( \alpha_g^{(t)} \nabla_\theta l(\pi_{\theta^{(t-1)}}; (x_g, y_w, y_l)) \big) \Big)$ // Use $\alpha$ to update $\theta$
10: **end for**
11: **Return:** Output the robust policy $\pi(\theta^{(T)})$

---

Next, we compute the derivative of $l(\theta; (x, y_w, y_l))$ w.r.t. $\theta$ from Equation (58) to check the Lipschitz continuity.

$$\nabla_\theta l(\theta; (x_g, y_w, y_l))$$
$$= \frac{\sigma\big(\beta\langle\phi(x_g, y_w) - \phi(x_g, y_l), \theta\rangle\big)\big(1 - \sigma(\beta\langle\phi(x_g, y_w) - \phi(x_g, y_l), \theta\rangle)\big) * (\phi(x_g, y_w) - \phi(x_g, y_l))}{\sigma\big(\beta\langle\phi(x_g, y_w) - \phi(x_g, y_l), \theta\rangle\big)}$$
$$= \Big(1 - \sigma\big(\beta\langle\phi(x_g, y_w) - \phi(x_g, y_l), \theta\rangle\big)\Big) * (\phi(x_g, y_w) - \phi(x_g, y_l))$$
$$= \sigma\big(\beta\langle\phi(x_g, y_w) - \phi(x_g, y_l), -\theta\rangle\big) * (\phi(x_g, y_w) - \phi(x_g, y_l)).$$

Then, the gradient norm can be bounded as follows:

$$\|\nabla_\theta l(\theta; (x_g, y_w, y_l))\| \leq \|\phi(x_g, y_w) - \phi(x_g, y_l)\|$$
$$\leq B_\phi.$$

Hence, by the definition of Lipschitz functions for continuously differentiable functions, we have that $l(\theta; (x, y_w, y_l))$ is Lipschitz continuous with Lipschitz constant $B_\phi$. Further, we bound the loss function using the bounds for $\theta$ and $\phi(\cdot, \cdot)$.

$$\langle\phi(x_g, y_w) - \phi(x_g, y_l), \theta\rangle \geq -\|\phi(x_g, y_w) - \phi(x_g, y_l)\|\|\theta\|$$
$$\langle\phi(x_g, y_w) - \phi(x_g, y_l), \theta\rangle \geq -B_\phi B_\Theta$$
$$\log(\sigma(\langle\phi(x_g, y_w) - \phi(x_g, y_l), \theta\rangle)) \geq \log(\sigma(-B_\phi B_\Theta))$$
$$-\log(\sigma(\langle\phi(x_g, y_w) - \phi(x_g, y_l), \theta\rangle)) \leq -\log(\sigma(-B_\phi B_\Theta))$$
$$\leq \log(1 + \exp(B_\phi B_\Theta))$$
$$\leq \log(\exp(B_\phi B_\Theta) + \exp(B_\phi B_\Theta))$$
$$\leq \log(2) + (B_\phi B_\Theta) = B_l$$

Hence, the loss function is bounded by $B_l$ for bounded $\theta$ and $\phi$.

$\square$

### E.2 Alternate Sampling Strategy

In this section, we study the convergence of an alternate algorithm in Algorithm 3, wherein groups are sampled uniformly rather than a categorical distribution proportional to the group sizes, in terms of the error $\epsilon_T$ as defined in Equation (46).

**Proposition E.2.** *Suppose that the loss $l(\cdot; (x_g, y, y'))$ is non-negative, convex, $B_\nabla-$Lipschitz continuous, and bounded by $B_l$ for all $(x_g, y, y') \in \mathcal{X} \oplus \mathcal{G} \times \mathcal{Y} \times \mathcal{Y}$ and $\|\theta\|_2 \leq B_\Theta$ for all $\theta \in \Theta$ with convex $\Theta \subset \mathbb{R}^d$. Then, the average iterate of Algorithm 3 achieves an error at the rate*

$$\epsilon_\Phi(\tilde{\zeta}^{1:T}) \leq 2\sqrt{10\frac{B_\Theta^2 B_\nabla^2 + K B_l^2 \ln K}{T}}. \tag{59}$$

*Proof.* We follow a similar proof structure as in the proof of Proposition 4.1. We recall the saddle-point stochastic optimization problem from [32] stated earlier in Equation (47),

$$\min_{\theta \in \Theta} \max_{\alpha \in \Delta_{K-1}} \left\{ \phi(\theta, \alpha) = \sum_{g=1}^{K} \alpha_g \mathbb{E}[F_g(\theta, \xi)] \right\}. \tag{60}$$

In lieu of the alternate sampling strategy in Algorithm 3 where groups are sampled uniformly, we consider,

$$\xi := (x_g, y, y') \sim \sum_{g=1}^{K} \frac{1}{K} \mathcal{D}_g, \tag{61}$$

and we define $F_{g'}(\theta, (x_g, y, y')) := K\mathbf{1}_{[g=g']} l(\theta; (x_g, y, y'))$. Then, the expectation

$$
\begin{aligned}
\mathbb{E}_\xi[F_g(\theta, \xi)] &= \sum_{g'=1}^{K} \frac{1}{K} \mathbb{E}_{\mathcal{D}_{g'}}[F_g(\theta, \xi)|g = g'] \\
&= \frac{1}{K} \mathbb{E}_{\mathcal{D}_g}[F_g(\theta, \xi)|g = g'] \\
&= \frac{1}{K} \mathbb{E}_{\mathcal{D}_g}[Kl(\theta; (x_g, y, y'))|g = g'] \\
&= \mathbb{E}_{\mathcal{D}_g}[l(\theta; (x_g, y, y'))|g = g'] \\
&= f_g(\theta)
\end{aligned}
$$

**Mirror Descent mapping.** We map Algorithm 3 to the general mirror descent framework of [32] as done in the proof of Proposition 4.1. We begin by recalculating the gradients of $\phi(\theta, \alpha)$ w.r.t. $\theta$ and $\alpha$ for this alternate definition of $F_g(\theta, \xi)$,

$$
\begin{aligned}
\partial\Phi(\theta, \alpha, \xi) &= \begin{bmatrix} \partial_\theta \sum_{g'=1}^{K} \alpha_{g'}[F_{g'}(\theta, \xi)] \\ -\partial_\alpha \sum_{g'=1}^{K} \alpha_{g'}[F_{g'}(\theta, \xi)] \end{bmatrix} \\
&= \begin{bmatrix} \partial_\theta \sum_{g'=1}^{K} \alpha_{g'}[K\mathbf{1}_{[g=g']} l(\theta; (x_g, y, y'))] \\ -\partial_\alpha \sum_{g'=1}^{K} \alpha_{g'}[K\mathbf{1}_{[g=g']} l(\theta; (x_g, y, y'))] \end{bmatrix} \\
&= \begin{bmatrix} K\alpha_g \nabla_\theta l(\theta; (x_g, y, y')) \\ -\left(0, \cdots, Kl(\theta; (x_g, y, y')), \cdots, 0\right) \end{bmatrix}.
\end{aligned}
$$

Based on the above gradient expressions, we can similarly map the updates of $\theta^t$ and $\alpha^t$ in Algorithm 3, to the general mirror descent update rule with corresponding prox-function as done for Proposition 4.1. We omit further details regarding the mirror descent related definitions as they follow from the proof of Proposition 4.1. We directly proceed to bounding the error $\epsilon_\phi(\tilde{\zeta}^{1:T})$ using [32, Equation 3.23],

$$\epsilon_\phi(\tilde{\zeta}^{1:T}) \leq 2\sqrt{10 \frac{R_\theta^2 M_\theta^2 + \ln(K)M_\alpha^2}{T}}. \tag{62}$$

We recalculate $M_\theta$ and $M_\alpha$ for this alternate definition of $F_g(\theta, \xi)$ as they correspond to the upper bounds to the maximum of the expected norms of $F_g(\theta, \xi)$ and $\nabla_\theta F_g(\theta, \xi)$ ,

$$
\begin{aligned}
\max_{1 \leq g \leq K} \mathbb{E}\|\nabla F_g(\theta, \xi)\|^2 = \max_{1 \leq g \leq K} \mathbb{E}\|K\mathbf{1}_{[g=g']} \nabla l(\theta; (x_g, y, y'))\|^2 &= \max_{1 \leq g \leq K} \frac{1}{K} K^2 \|\nabla l(\theta; (x_g, y, y'))\|^2 \\
&= \max_{1 \leq g \leq K} K\|\nabla l(\theta; (x_g, y, y'))\|^2 \\
&\leq B_\nabla^2 K \\
&= B_\nabla^2 K \\
&= M_\theta^2
\end{aligned}
$$

Similarly,

$$\mathbb{E} \max_{1 \le g \le K} \|F_g(\theta, \xi)\|^2 = \mathbb{E} \max_{1 \le g \le K} \|K\mathbf{1}_{[g=g']} l(\theta; (x_g, y, y'))\|^2 \le \sum_{g=1}^{K} \frac{1}{K} K^2 \|l(\theta; (x_g, y, y'))\|^2$$

$$= \sum_{g=1}^{K} K \|l(\theta; (x_g, y, y'))\|^2$$

$$\le B_l^2 K^2$$

$$= B_l^2 K^2$$

$$= M_\alpha^2$$

Here, we have used $\|\nabla_\theta l(\theta; (x_g, y))\| \le B_\nabla$ and $\|l(\theta; (x_g, y))\| \le B_l$.

Using the alternate $M_\theta$ and $M_\alpha$ corresponding to this sampling rule, we obtain the overall error bound as follows:

$$\epsilon_\Phi(\tilde{\zeta}^{1:T}) \le 2\sqrt{10K \frac{B_\Theta^2 B_\nabla^2 + B_l^2 K \ln K}{T}}. \tag{63}$$

$\square$

