# OpenReview forum: "Group Robust Preference Optimization in Reward-free RLHF"
_NeurIPS.cc/2024/Conference — NeurIPS 2024 poster_

### Official Review · Reviewer_SYLq · 2024-07-11

**Soundness:** 3
**Presentation:** 3
**Contribution:** 2
**Rating:** 4
**Confidence:** 4

**Summary:**

This paper proposes a novel preference optimization technique, GRPO, which utilizes the group distributional robust optimization. Specifically, the method aims to maximize the worst-case group performance for improving the robustness of LLM models. This paper provides several theoretical aspects of GRPO when it comes to convergence guarantee and closed-form solution when considering the log-linear policy class. Empirically, the GRPO objective function is minimized by the mirror gradient descent. Several theoretical results on synthetic and real-world datasets demonstrate that their method works well.

**Strengths:**

* This is the first study to adopt the group DRO technique in the RLHF setting.

* Several theoretical results are provided to enhance understanding of the proposed method.

* The proposed method is general so that it can be applied to various RLHF optimization techniques.

**Weaknesses:**

* This paper lacks novelty in its contribution. The group DRO technique is already a popular method used in a variety of applications [1, 2].  While the authors offer some theoretical results and an algorithm for their method, these may also lack novelty. In specific, Proposition 3.1 on the Nash equilibrium is a well-known result in game theory and Proposition 3.2 seems straightforward because $\pi^*$ is dependent only on an input value, not on the group to which the input belongs. Furthermore, there is no significant difference in the algorithm and proof of convergence guarantee compared to existing works [37, 30 in the manuscript], except for slight modifications.

* Although this paper is novel in applying the group DRO technique to RLHF preference optimization for the first time, stronger motivations and empirical results are needed to highlight the necessity of the proposed method. But, I also think that to strengthen this contribution, For example, as mentioned in lines 141-153, it would be beneficial to show that existing Large Language Models (LLMs) suffer from group robustness issues in various applications involving helpful/harmful instances or domain-specific categories when fine-tuned by RLHF optimization techniques.

* The interpretation of the results in Section 5.2 is insufficient. In Figure 3, GR-IPO improved the performance of all groups, including Group 1. This result contrasts with findings reported in the Group DRO paper [37], where a trade-off between majority and minority group performance is generally observed, with majority group performance typically decreasing to compensate for minority group performance. Therefore, a more thorough analysis or performance comparison is needed to explain this result.

[1] Re-weighting based group fairness regularization via classwise robust optimization, 2023.
[2] Distributionally Robust Multilingual Machine Translation, 2021.

**Questions:**

1. In the experiments in Section 5.2, what are the zero-shot performances of Gemma-2B for each group? It would be better to report the degree of bias in the pre-trained model together, because it can enable comparing the bias degrees between the pre-trained and fine-tuned models.

2. Would you provide the results when fine-tuning more layers or the entire neural networks using your method? I think that this can strengthen this paper's contribution by showing the performance of the method in various settings.

**Limitations:**

They addressed the limitations of this paper in the manuscript.

---

> ### Author Rebuttal · Authors · 2024-08-07
>
> We thank the reviewer for highlighting the novel application of our technique to the RLHF setting, our insightful theoretical analysis, and the broad applicability of our proposed algorithm.
>
> Next, we provide answers to **all** the questions posed by the reviewer.
>
> **Regarding theoretical novelty:**
>
> We appreciate the reviewer's evaluation of the theoretical results in our paper, though some key aspects of our contributions may have been overlooked.
>
> We agree that the group DRO is widely used, as noted in Lines 56-60. However, to the best of our knowledge, we are the first to apply it for LLM alignment. Specifically, we motivate and formulate the first group robust alignment of LLMs and derive a novel group robust MLE objective (Eq. 6). Additionally, we show that a naive application of group robustness to the LLM policy maximization objective does not offer robustness benefits (Lemma B.2 in Appendix B).
>
> Regarding our analysis, while Proposition 3.1 is well-known in game theory, it guarantees the existence of Nash equilibria for the special case of log-linear policy classes and supports our algorithmic design. Compared to [37,30], we adopt an alternate sampling strategy (see Line-5 of Algorithm-1), which samples groups proportionally to their sizes and assigns uniform weights to all the data points. From a practical perspective, this facilitates stable multi-epoch batch training. However, this might cause samples from smaller groups to be sparsely included in a given batch, thus requiring a compensatory scaling factor of $N/N_g$ in the update of Lines 6 and 8 in Algorithm-1. As a result, our algorithm requires a different proof technique compared to [37,30] in order to attain the convergence guarantees as detailed in Appendix C.
>
> Furthermore, our development of the GR-IPO objective for the log-linear policy class yields a closed-form weighted regression update for the policy parameters rather than a gradient update (Section 4.1 and Appendix C). To the best of our knowledge, this is a novel contribution towards efficient fine-tuning through preferential data. We are happy to further elaborate on these differences in our paper.
>
> **Regarding other potential use cases:**  In Lines 141 - 153, we outline additional potential applications of our approach, to engage readers and inspire future research. Our primary focus is on applying our approach to **pluralistic alignment tasks**, such as ensuring equitable alignment across diverse demographic preferences, as explored in related works (e.g., [1,2]).  While extending our method to **multi-objective** applications is interesting, it falls outside the intended scope of this work (and requires additional computational resources), which we leave for future research.
>
> Regarding helpful and harmful instances, we refer the reviewer to [3], which discuss the trade-offs between optimizing for helpfulness and minimizing harmfulness (Section 5.1) in LLM fine-tuning. Unlike their approach, which uses an additional hyperparameter to control the trade-offs, our approach is “parameter-free”, ensuring equitable performance and proving more effective for multi-objective tasks.
>
> **Regarding improved performance across groups compared to non-robust baselines:**
>
> We thank the reviewer for their detailed examination of our results. We agree with the reviewer that GRPO improves the worst group's performance, often reducing the average or best group's performance. In contrast, we can observe improved performance across all groups (see Figure-3 top right of our paper). This is only due to the **erroneous inclusion** of an IPO run with zero training steps while plotting the IPO group losses. The corrected plots are now in the attached PDF (Figure R3), where the new plot shows Group-5 loss of IPO and max group loss of IPO (Figure-3 top left) indeed align, unlike in the previous version. The performance figures for GR-IPO were already accurate and unchanged between the old and new plots. With this revised plot, we note that our method, GR-IPO, improves Group 5's performance, albeit at the expense of Group 1 which is consistent with the reviewer’s observations.
>
> **Regarding zero-shot performances of pre-trained Gemma-2B model:**
>
> We have included the zero-shot performance of the Gemma-2B model in Figure R4 of the attached PDF as requested. Significant group bias is observed in the pre-trained model's performance. However, we would like to emphasize that it is more relevant to view the performances of SFT fine-tuned model as our robust methodology builds upon it. We visualize the SFT performance (on the same data) in Figure 3 (Bottom right in the paper) and note that the SFT model’s degree of group bias is aligned to those of weights and group losses in Figure 3 (Bottom middle and top right). For further comparisons, we kindly refer to Lines (312-319) and our response to Reviewer kmNK on the alignment performance of the fine-tuned vs. base model.
>
> **Regarding fine-tuning more layers:**
>
> Thanks for the question. We believe there has been a misunderstanding here. Although our theoretical framework considers only the last layer fine-tuning, in our experiments, we apply the LoRA strategy to fine-tune **all layers of** the model as detailed in the code (also available). To avoid further confusion, we will clearly state this comprehensive fine-tuning approach in our experiments and give details in the Appendix.
>
> We have addressed all the questions raised by the reviewer. In light of the reviewer’s opinions about the strengths of our work and our detailed rebuttal to the reviewer’s questions, we kindly ask the reviewer to reconsider their score.
>
> [1] Zhao, Siyan et al. "Group preference optimization: Few-shot alignment of large language models."
> [2] Sorensen, Taylor, et al. "A roadmap to pluralistic alignment."
> [3] Bai, Yuntao, et al. "Training a helpful and harmless assistant with reinforcement learning from human feedback."

---

### Official Review · Reviewer_7Bpj · 2024-07-13

**Soundness:** 3
**Presentation:** 2
**Contribution:** 3
**Rating:** 5
**Confidence:** 4

**Summary:**

This paper addresses the limitation of traditional reinforcement learning with human feedback (RLHF) approaches that indiscriminately optimize a single preference model, disregarding the unique characteristics and needs of diverse labeler groups. The authors propose a Group Robust Preference Optimization (GRPO) method to align LLMs with individual groups' preferences.

**Strengths:**

The paper's motivation is clear.

**Weaknesses:**

1. How does the proposed method handle scenarios where preference data lacks clear group information? In practice, much preference data does not come with explicit group classifications.

2. The paper should address whether the proposed method can learn an invariant preference standard from data that may contain various group information, such as different preference evaluation criteria and preferences annotated at different iterations.

3. The authors need to provide and analyze the alignment results of the model, not just the reward model analysis.

**Questions:**

Please refer to the Weaknesses section for questions regarding the paper.

**Limitations:**

Yes

---

> ### Author Rebuttal · Authors · 2024-08-07
>
> We thank the reviewer for their positive outlook on our work and for highlighting the clear motivation of our problem setting to address the limitations in traditional RLHF techniques.
>
> **Regarding access to group information:**
>
> We focus on settings with known groups, which are common in pluralistic alignment datasets and tasks (see [1,2]). When groups are unknown, one can still apply several approaches that we hypothesize below.
>
> To identify unknown groups in data, one can: a) use clustering: Apply algorithms like k-means or EM to detect potential groups; b) apply representation learning: Use techniques like variational autoencoders (VAEs) to discover hidden structures; c) analyze features: Look for features with high variance or correlations to hypothesize groupings; d) consult experts: Leverage domain expertise to identify meaningful groups. These methods can help uncover group structures in the data.
> We believe that this is not a limitation of our work and can be an interesting direction for future research.
>
> **Regarding learning an invariant preference standard from data that may contain various group information:**
>
> We kindly ask the reviewer to clarify what an invariant preference standard means in this context so that we can provide an appropriate answer to this question.
>
> **Regarding the analysis of alignment results of the model:**
>
> We would like to refer the reviewer to Figure-3 (Bottom left), where we indeed analyze the model’s alignment through worst group log-prob. accuracies. The log-prob. accuracy measures the accuracy of the model to assign higher probability to the chosen response compared to the rejected response (see Lines 302-304). We note that the GR-IPO fine-tuned model is better aligned with the worst group compared to the IPO fine-tuned model.
>
> Further, we plot and compare the alignment performance of GR-IPO, IPO, and SFT fine-tuned models with the pre-trained model across groups in Figure R2 of the attached PDF. Note that the pre-trained model performs significantly worse in comparison to other methods and attains its worst performance in Group 4. Further, GR-IPO outperforms both IPO and SFT in terms of worst group performance (Group 5). This is consistent with our methodology that aims to improve the alignment of the LLM and reduce the bias across groups.
>
> We believe that we have answered all the questions raised by the reviewer. In light of the reviewer’s opinions about the strengths of our work and our detailed rebuttal to the reviewer’s questions, we kindly ask the reviewer to reconsider their score. We are happy to answer and clarify any further questions the reviewer raises.
>
> [1]. Zhao, Siyan, John Dang, and Aditya Grover. "Group preference optimization: Few-shot alignment of large language models."
> [2] Sorensen, Taylor, et al. "A roadmap to pluralistic alignment." arXiv preprint arXiv:2402.05070 (2024).

---

### Official Review · Reviewer_kmNK · 2024-07-13

**Soundness:** 3
**Presentation:** 3
**Contribution:** 3
**Rating:** 7
**Confidence:** 3

**Summary:**

The work tackles the important problem of robust RLHF for diverse groups. Traditionally, RLHF assumes that a single model can fit the diverse feedback from multiple groups of users. In this paper, the authors introduce a method to learn a robust policy that maximizes for worst-case group performance. To achieve this, the method adaptively weights the loss for each group based on the size and cumulative training loss incurred by the feedback samples for that group. As a result, LLMs trained on diverse group data demonstrate reduced loss imbalance and improved accuracies across all the groups. The authors also present a convergence analysis of the proposed method assuming a log-linear policy class.

**Strengths:**

1. The paper tackles a critical problem of robust RLHF for diverse groups. The intuition of the overall method is well understood, the framework and the parameters are clearly mentioned, and the results are shown over the standard datasets and compared to multiple baselines.
2. The authors present a thorough literature review and background. Past work has mainly focused on making RLHF and LLMs robust to noisy or out of distribution data. Meanwhile, this work focuses on a group robust formulation of training LLMs using state-of-the-art methods (mainly DPO).
3. The method introduced “GRPO” is useful is scenarios beyond diverse groups. As the authors mention, it is a general formulation that can enforce robustness to diverse tasks, domains, or objectives occurring in the feedback dataset.
4. To achieve robustness, GRPO presents a robust optimization approach to minimize worst-case loss amongst the diverse groups. Further, the paper introduces a less aggressive object by trading off worst and average case performance. I would be curious to see an ablation study showing the effectiveness of this tradeoff.
5. The paper supplements the approach with some strong theoretical proofs on the convergence properties of the method under a log-linear policy. The authors also present a close form solution for the RLHF update step, replacing DPO with IPO.
6. The results show that GRPO improved performance across all groups, and the weight update behaves as expected by assigning higher weights to groups with higher cumulative loss i.e. the gradients from the worst performing group are scaled the most.

**Weaknesses:**

1. This paper proposes a method for group robust optimization for LLMs. However, the metrics evaluated are only max validation loss and reward error over the groups. GRPO uses a reward free approach to update the LLM, but the evaluations are restricted to the performance of the reward model over the feedback dataset. It would be nice to see the performance of the finetuned vs base model (such as win rate) in generating responses that align with the individual groups.
2. The authors provide detailed training setup, but I would suggest that they also include information about the evaluation.
3. GRPO performs well over all the groups, however, the performance of the importance sampling baseline is very close. It would be helpful if the authors could provide additional ablations and discussions to show the effectiveness of optimizing against the worst-case loss over the groups vs using only the IS approach.

**Questions:**

1. In the current experiments, is only the final layer of the LLM trained? I would be curious to know the result if the method used full finetuning of the model.
2. As GRPO assumes that each prompt has access to group information through a prompt, how much does the prompt affect policy? If the prompts are good enough, would it just create non-overlapping distributions for all groups? As one of the problems in non-robust baselines is that they converge to unwanted biases or majority groups for shared prompts, if the group identification prompt is finetuned will it alleviate this issue altogether?
3. Does GRPO finetune only the final layer of the Gemma-2B model in the results presented in the paper?
4. Does GRPO ensure that there is a non-decreasing change in performance across all groups as compared to the non-robust baselines?

**Limitations:**

1. The paper introduces GRPO, an optimization method to weight each group in the RLHF update step proportional to the size and loss of the group. This ensures a balanced performance of the model across all the groups. In the results, we see an improved performance over all the individual groups, which intuitively violates the no-free lunch theorem. I would be curious to see an analysis of the method that gives insights as to what allows the model or the objective to achieve consistently higher performance.
2. Here, the method assumes access the groups in the dataset. However, in practical settings the group information is unavailable and the model has to cluster or implicitly model the group information from the ungrouped dataset.
3. The evaluation is limited only to the accuracy of the reward model over the preference dataset. So, currently, it provides weaker evidence of the translation of this robustness during the generation phase. It would be nice if the authors could include experiments showing if GRPO enables LLMs to robustly generate better-aligned responses to prompts from all the groups.

---

> ### Author Rebuttal · Authors · 2024-08-07
>
> We thank the reviewer for their positive opinion about our work recognizing the importance of our problem setting,  broad applicability of our GRPO approach, and strong theoretical analysis of our proposed algorithm’s convergence properties.
>
> **Regarding ablation for trade-off parameter between worst-case and average performance:**
>
> Due to time constraints, we provide the results of our ablation study for the synthetic experiments in Figure R1 of the attached PDF. Note that as observed in Figure R1 of the attached PDF, the max validation loss decreases while moving from $\chi=0$ to $\chi=1$, where $\chi=0$ corresponds to importance sampling with group weights $\mu_1,\cdots,\mu_g$ mapping to importance sampling weights (see Eq. [30] in Appendix B.4) and $\chi=1$ corresponds to GR-IPO. Further, we plot the average validation loss which increases while moving from $\chi=0$ to $\chi=1$, demonstrating the trade-off between average and worst-case performance. Note that, GR-IPO aptly increases the average loss (as expected) in order to reduce the worst group loss.
>
> **Regarding details about evaluation:**
>
> Our primary evaluation metrics are the worst group loss and accuracy. The loss refers to the IPO loss for each group and the accuracy refers to the percentage of winning response and losing response pairs correctly ordered by the learned preference function [Eq. 35 in Appendix]. We have defined this in the main text in Lines 302-304 and also in Appendix D.4 (Lines 683-684) along with details about data splits for training, test, and validation (Lines 672-673). We will revise and emphasize the definitions in a more visible manner.
>
> **Regarding alignment performance of finetuned vs base model and alternate evaluation metrics:**
>
> We would like to refer the reviewer to Figure-3 (Bottom left), where we indeed measure the model’s alignment through worst group log-prob. accuracies. The log-prob. accuracy measures the accuracy of the model to assign higher probability to the chosen response compared to the rejected response (see Lines 302-304). We note that the GR-IPO fine-tuned model is better aligned with the worst group compared to the IPO fine-tuned model.
>
> Moreover, in our experiments, the responses/choices are included in the prompt and, our goal is to measure whether the chosen choice (for e.g., A) is preferred over the rejected choice (for e.g., C). Hence we focus on log-prob. accuracies metric. Further, as per the reviewer’s request, we plot and compare the alignment performance of GR-IPO, IPO, and SFT fine-tuned models with the pre-trained model across groups in Figure-R2 of the attached PDF. Note that the pre-trained model performs significantly worse in comparison to other methods and attains its worst performance in Group 4. And, GR-IPO outperforms both IPO and SFT in terms of worst group performance (Group 5). This is consistent with our methodology that aims to improve the alignment of the LLM and reduce the bias across groups w.r.t. SFT fine-tuned model rather than the pre-trained model.
>
> **Regarding proximity between the performances of importance sampling and GRPO:**
>
> We kindly request the reviewer to clarify the particular figure to which they are referring to. We note that in the two scenarios corresponding to Figure 2 and Figure 5 (Appendix D.2), the groups have different responses’ distributions and there is a clear and significant gap between our proposed method and the importance sampling approach.
> However, we agree that in Figure-4 of Appendix D.2, the gap between GR-IPO/GR-DPO and the corresponding importance sampling methods is indeed small. This is because Figure 4 corresponds to the scenario where both groups have the same responses’ distribution but are imbalanced in size. In this scenario, such a small performance gap is expected considering the difference between groups arises solely from data imbalance, which is handled by importance sampling.
>
> **Regarding fine-tuning more layers:**
>
> We kindly refer the reviewer to our response to reviewer SYLq regarding the same question
>
> **Regarding prompt-tuning techniques to alleviate group biases:**
>
> We agree that the prompt does affect the policy response and policy optimization, as studied in various previous works, including [2]. In our experiments, the prompts are appended with group information (detailed in Lines 669-670 of Appendix D.4), creating non-overlapping distributions for all groups. However, we disagree with the reviewer that tuning prompts alone will alleviate the issue of biased performance across groups.  Even with tuned group identification prompts, all groups will still use the same prompt template with only the group information varying. Hence, there is no guarantee that the IPO-based fine-tuning will lead to reduced bias across groups, as the losses might still be distributed unevenly as we observed in the performances of both SFT and IPO fine-tuned models (see Figure-3). Therefore, a group robust fine-tuning strategy like GRPO is still necessary to reduce bias across groups.
>
> **Regarding improved performance across groups compared to non-robust baselines:**
>
> We kindly refer the reviewer to our response to reviewer SYLq regarding the same question
>
> **Regarding access to group information:**
>
> We kindly refer the reviewer to our response to reviewer qNjA regarding the same question. We believe that this is not a limitation of our work and can be an interesting direction for future research.
>
> [1] Lester, Brian, Rami Al-Rfou, and Noah Constant. "The power of scale for parameter-efficient prompt tuning." arXiv preprint arXiv:2104.08691 (2021).
> [2] Hu, Edward J., et al. "Lora: Low-rank adaptation of large language models." arXiv preprint arXiv:2106.09685 (2021).

---

> > ### Comment · Reviewer_kmNK · 2024-08-11
> >
> > I thank the reviewers for a detailed response to my concerns. I have read the rebuttals, and it adequately addresses all my questions. I am increasing the score to accept.

---

> > > ### Author Response · Authors · 2024-08-12
> > >
> > > We thank the reviewer for acknowledging our detailed rebuttal and raising their score.

---

### Official Review · Reviewer_qNjA · 2024-07-16

**Soundness:** 3
**Presentation:** 3
**Contribution:** 3
**Rating:** 6
**Confidence:** 3

**Summary:**

This paper introduces GRPO, a method to optimize policy preferences across different groups in a robust way. GRPO looks at the worst group alignment loss by taking the maximum loss across all groups, ensuring the policy performs well even when there are group-specific differences or overlaps in prompts.

**Strengths:**

1. This paper writes well and easy to follow.
2. The paper provides a thorough theoretical analysis of GRPO.
3. The proposed framework accommodates both distinct and overlapping group scenarios.
4. GRPO can trade off worst-case for average performance with a hyperparameter.

**Weaknesses:**

1. GRPO's experiment on real world datasets is limited, on only one group (5 countries) of the GlobalQA dataset.
2. How is the training time as compared to other baselines? The GRPO framework's training process involves a min-max optimization, which can be potentially computationally intensive.

**Questions:**

please see weakness.

**Limitations:**

1. This methods requires knowing the number and nature of groups in advance to perform optimization.
2. When new group is introduced, will have to train the entire optimization again, which is expensive.

---

> ### Author Rebuttal · Authors · 2024-08-07
>
> We thank the reviewer for their positive opinion about our work recognizing our thorough theoretical analysis and broad applicability of our GRPO approach.
>
> **Regarding the groups in GlobalQA dataset experiment:**
>
> Our proposed GRPO method can be applied to any set of finite groups. To demonstrate the effectiveness of our method, we focus on five groups that are diverse in terms of both dataset size and ethnicity (see Lines 295-296).
>
> **Regarding training time comparison:**
>
> The training time for both IPO and GR-IPO are *almost the same* and are approximately around 4-5 hours for the GlobalOpinionQA data run on pre-trained Gemma-2B model. We have detailed the time and the configuration of the GPU processors used for our experiments in Appendix D.4 (Lines 685-688).
>
> Explanation: We do not need to solve the inner maximization in our minimax problem (see Eq. [6]) at each iteration. Instead, we update the weights over the groups based on the loss of the current sample as per Line 6 of our algorithm. Thanks to Danskin’s theorem, computing the gradient over the policy parameter $\theta$ only requires computing the partial gradient w.r.t. $\theta$ for a fixed $\alpha$, which approximates the maximizer $\alpha^*(\theta)$. As $\alpha$ is also updated iteratively, the computation time is comparable to the amount of computation time in the standard case. Thus, it does not lead to any significant computational overhead.
>
> **Regarding the requirement of knowing the number of groups in advance:**
>
> We focus on settings with known groups, which are common in pluralistic alignment datasets and tasks (see [1,2]). When groups are unknown, one can still apply several approaches that we hypothesize below.
>
> To identify unknown groups in data, one can: a) use clustering: Apply algorithms like k-means or EM to detect potential groups; b) apply representation learning: Use techniques like variational autoencoders (VAEs) to discover hidden structures; c) analyze features: Look for features with high variance or correlations to hypothesize groupings; d) consult experts: Leverage domain expertise to identify meaningful groups. These methods can help uncover group structures in the data.
>
> **Regarding requirement of re-training when new group is introduced:**
>
> We agree with the reviewer that introducing a new group requires additional training. However, note that one does not need to restart from scratch. The current policy parameter serves as a warm starting initialization point. Moreover, such additional training is necessary to ensure equitable alignment for both new and existing groups. We do not consider this an explicit limitation of our proposed algorithm, but rather an interesting future extension of our approach to continual learning settings.
>
> [1]. Zhao, Siyan, John Dang, and Aditya Grover. "Group preference optimization: Few-shot alignment of large language models."
> [2] Sorensen, Taylor, et al. "A roadmap to pluralistic alignment." arXiv preprint arXiv:2402.05070 (2024).

---

### Official Review · Reviewer_V78J · 2024-07-17

**Soundness:** 3
**Presentation:** 3
**Contribution:** 3
**Rating:** 6
**Confidence:** 2

**Summary:**

This paper addresses the problem of improving fairness in preference optimization. It proposes a new loss function and algorithm. Experiments conducted show that the proposed algorithm indeed achieves better fairness. Additionally, the paper provides a theoretical analysis indicating the convergence of the optimization.

**Strengths:**

1. The idea of improving the fairness is good. This is a problem that deserves more attention.

2. The derivation of the loss and the algorithm, though simple, is clear.

3. The results provided for the algorithm appear to be sound.

**Weaknesses:**

1. The author mentioned that the idea can also be applied to multi-objective preference optimization. In my opinion, a comparison with works in multi-objective preference optimization [1] and group optimization [2] is needed.



2. It is unclear whether this algorithm can be applied to scenarios beyond multiple choices.

3. It is unclear how Proposition 4.1 relates to common metrics like regret.


[1]. Zhao, Siyan, John Dang, and Aditya Grover. "Group preference optimization: Few-shot alignment of large language models."

[2]. Wang, Haoxiang, et al. "Arithmetic control of llms for diverse user preferences: Directional preference alignment with multi-objective rewards."

**Questions:**

1. The paper mentioned that the algorithms can also be used for multi-objective preference optimization. Were any related experiments conducted?

**Limitations:**

See the 'Weakness' part

---

> ### Author Rebuttal · Authors · 2024-08-07
>
> We thank the reviewer for their positive opinion about our work recognizing the importance of the problem, clear and sound exposition of our algorithm and theoretical results.
>
> **Regarding comparison with works in multi-objective preference optimization [1] and group optimization [2] and related experiments:**
>
> The primary focus of this paper is to design a robust fine-tuning approach to align a language model with diverse group preferences and validate it both theoretically and numerically. Indeed, our method applies to robust multi-objective preference optimization. It is definitely interesting to conduct experiments for the robust multi-objective setting. However, these are beyond the scope of this project and will be considered for future work.
>
> Compared to [1], the distinctive feature of our method is that we are not modeling multi-reward objectives but consider a reward-free setting. Specifically, we consider the robust alignment problem optimizing for the worst group performance. Further, they align with user preferences assuming that each user/group has varied importance over the distinct metrics in their multi-objective reward model. The output policy is trained to output a response based on both the prompt and the importance/weights over the individual metrics. Whereas, our methodology directly models each group’s preferences through a group-dependent latent reward model where the group dependency is injected through the prompt.
>
> Compared to [2], as explained in Lines 68-71 and also in the appendix (Lines 520-523), we note that they consider alignment to multiple groups’ preferences through an in-context learning approach that is different from our LLM fine-tuning methodology. In particular, they consider a distributional objective that is different from the robust one we consider.
> Hence, we believe that [1,2] are not directly comparable to ours and we will include this explanation and comparison regarding [1] in our related work section.
>
> **Regarding application of our algorithm to scenarios beyond multiple choices:**
>
> Yes, it can be used for scenarios beyond multiple choices. The reason why we use the global opinion data with a multiple choice structure in our experiments is to clearly demonstrate the efficacy of our approach in aligning LLMs to diverse group preferences and such datasets are typically used in similar pluralistic alignment studies (e.g., [2,3]).
>
> **Regarding relation of Proposition 4.1 with regret:**
>
> Proposition 4.1 is indeed related to regret. It bounds the expected difference in loss between the average iterate policy and the optimal policy after T iterations and demonstrates a sublinear dependency on T. This allows us to provide convergence guarantees for our algorithm in terms of the average iterate. Such a regret formulation is common in minimax problems as discussed in Nemirovski et al. [30].
>
> [1] Wang, Haoxiang, et al. "Arithmetic control of llms for diverse user preferences: Directional preference alignment with multi-objective rewards."
> [2] Zhao, Siyan, John Dang, and Aditya Grover. "Group preference optimization: Few-shot alignment of large language models."
> [3] Sorensen, Taylor, et al. "A roadmap to pluralistic alignment." arXiv preprint arXiv:2402.05070 (2024).

---

### Author Rebuttal · Authors · 2024-08-07

We thank the reviewers for their valuable feedback and for recognizing the strengths of our work. We appreciate the constructive comments raised by the reviewers and we believe we have addressed all of them in detail further strengthening the validity of our work.

In summary, the reviewers recognize the following **strengths** of our work\
(i) The criticality of our problem setting in the current LLM alignment paradigm\
(ii) The broad applicability of our approach in the real-world\
(iii) Strong theoretical backing of our methodology and algorithm\
(iv) Consistent experimental setup showcasing the efficacy of our approach

Further, the following shared comments were mentioned by reviewers:

**(i) Regarding the requirement of knowing the number of groups in advance:** \
We agree with the reviewers that our method requires knowing the groups in advance and note that we focus on settings with known groups, which are common in pluralistic alignment datasets and tasks (see [1,2]). Hence, we do not see this as an explicit limitation of our approach. We also provide details on other approaches that can be employed to learn the groups, when they are unknown, in our responses to reviewers qNjA and 7Bpj.

**(ii) Regarding alignment performance of finetuned vs base model:** \
As per the reviewers’ requests, we plot and compare the alignment performance of GR-IPO, IPO, and SFT fine-tuned models with the pre-trained model across groups in Figure R2 of the attached PDF. Note that the pre-trained model performs significantly worse in comparison to other methods and, our approach GR-IPO outperforms both IPO and SFT in terms of worst group performance.
We also provide further details about the plot in our responses to reviewers kmNK and 7Bpj.

**(iii) Regarding fine-tuning more layers:** \
Although our theoretical framework considers only the last layer fine-tuning, in our experiments, we apply the LoRA strategy to fine-tune all layers of the model as detailed in the code (also available). We will clearly state this comprehensive fine-tuning approach in our experiments and give details in the Appendix.

Further, as per the requests of reviewers,\
(i)  We have included the zero-shot performance of the Gemma-2B model in Figure R4 of the attached PDF
(ii) We provide the results of our ablation study for the synthetic experiments in Figure R1 of the attached PDF

We have also responded to each reviewer’s questions and comments individually. We believe we have addressed all of them thoroughly and are happy to answer any further questions that are raised by the reviewers.

[1]. Zhao, Siyan, John Dang, and Aditya Grover. "Group preference optimization: Few-shot alignment of large language models." \
[2] Sorensen, Taylor, et al. "A roadmap to pluralistic alignment." arXiv preprint arXiv:2402.05070 (2024).

---

### Author Response · Authors · 2024-08-12
**Gentle Reminder**

We thank all the reviewers again for the time taken to review our work. As the author-reviewer discussion period is nearing its end, we would appreciate it if the reviewers would respond to our rebuttals. If there any further questions, we are more than happy to clarify and discuss further.

---

### Decision · Program_Chairs · 2024-09-25

**Decision:**

Accept (poster)

**Comment:**

This paper brings attention to the existence of diverse preferences across different groups, proposing a robust RLHF method. Defining the optimization problem with the weighted loss for each group, the authors have discovered that the problem can be reduced to a minimax optimization problem. The minimax optimization problem is a convex-concave game, and by combining MWU and gradient descent updates, it can be guaranteed from classical results that the average iterate will converge to the equilibrium. When the policy is log-linear, essentially one can compute the best response of one player each round analytically (as a classical regression), resulting in a simplified algorithm. Through synthetic experiments and real-world data experiments, the authors demonstrate that their method performs best for the group that would likely have the worst loss, suggesting the significance of the approach in the real world. There was a concern from a reviewer about other applications and motivations, and questions about improved performance seen across all groups (in fact, the authors included an incorrect plot in the initial manuscript). In their rebuttal, the authors presented the correct experimental results, which were confirmed to be consistent. Overall, while the technical/experimental contributions are not particularly significant, the contribution of bringing the concept of group robustness to LLM alignment and conducting appropriate analysis is substantial, which is generally recognized by the reviewers.